# Implicit Sparse Regularization:
# The Impact of Depth and Early Stopping

**Jiangyuan Li**
jiangyuanli@tamu.edu
Texas A&M University

**Thanh V. Nguyen**
thanhng.cs@gmail.com
AWS AI

**Chinmay Hegde**
chinmay.h@nyu.edu
New York University

**Raymond K. W. Wong**
raywong@tamu.edu
Texas A&M University

## Abstract

In this paper, we study the implicit bias of gradient descent for sparse regression. We extend results on regression with quadratic parametrization, which amounts to depth-2 diagonal linear networks, to more general depth-$N$ networks, under more realistic settings of noise and correlated designs. We show that early stopping is crucial for gradient descent to converge to a sparse model, a phenomenon that we call *implicit sparse regularization*. This result is in sharp contrast to known results for noiseless and uncorrelated-design cases. We characterize the impact of depth and early stopping and show that for a general depth parameter $N$, gradient descent with early stopping achieves minimax optimal sparse recovery with sufficiently small initialization $w_0$ and step size $\eta$. In particular, we show that increasing depth enlarges the scale of working initialization and the early-stopping window so that this implicit sparse regularization effect is more likely to take place.

## 1 Introduction

**Motivation.** Central to recent research in learning theory is the insight that the choice of optimization algorithms plays an important role in model generalization [1, 2, 3]. A widely adopted view is that (stochastic) gradient descent — the most popular optimization algorithm in machine learning — exhibits some implicit form of regularization. Indeed for example, in the classical under-determined least squares setting, gradient descent (with small step size) starting from the origin converges to the model with minimum Euclidean norm. Similar implicit biases are also observed in deep neural network training in which the networks typically have many more parameters than the sample size. There, gradient descent without explicit regularization finds solutions that not only interpolate the training data points but also generalize well on test sets [2, 4, 5, 6, 7].

This insight, combined with the empirical success stories of deep learning, has sparked significant interest among theoretical researchers to rigorously understand implicit regularization. The majority of theoretical results focus on well-understood problems such as regression with linear models [8, 9, 10, 11, 12, 13] and matrix factorization [14, 15, 16, 17], and show that the parametrization (or architecture) of the model plays a crucial role. For the latter, Gunasekar et al. [15] conjectured that gradient descent on factorized matrix representations converges to the solution with minimum nuclear norm. The conjecture was partially proved by Li et al. [16] under the Restricted Isometry Property (RIP) and the absence of noise. Arora et al. [17] further show the same nuclear-norm implicit bias using depth-$N$ linear networks (i.e., the matrix variable is factorized into $N$ components).

35th Conference on Neural Information Processing Systems (NeurIPS 2021).

Parallel work on nonlinear models and classification [5, 9] has shown that gradient descent biases the solution towards the max-margin/minimum $\ell_2$-norm solutions over separable data. The scale of initialization in gradient descent leads to two learning regimes (dubbed "kernel" and "rich") in linear networks [18], shallow ReLU networks [19] and deep linear classifiers [20]. Li et al. [21] showed that depth-2 network requires an exponentially small initialization, whereas depth-$N$ network ($N \geq 3$) only requires a polynomial small initialization, to obtain low-rank solution in matrix factorisation. Woodworth et al. [18] obtained a similar result for high dimensional sparse regression.

The trend in the large majority of the above works has been to capture implicit regularization of gradient descent using some type of norm with respect to the working parametrization [18, 22, 23, 24]. On the other hand, progress on understanding the *trajectory* of gradient descent has been somewhat more modest. [11, 12] study the sparse regression problem using quadratic and Hadamard parametrization respectively and show that gradient descent with small initialization and careful *early stopping* achieves minimax optimal rates for sparse recovery. Unlike [16, 18] that study noiseless settings and require no early stopping, [11, 12] mathematically characterize the role of early stopping and empirically show that it may be necessary to prevent gradient descent from over-fitting to the noise. These works suggest that the inductive bias endowed by gradient descent may be influenced not only by the choice of parametrization, *but also algorithmic choices* such as initialization, learning rate, and the number of iterations. However, our understanding of such gradient dynamics is incomplete, *particularly* in the context of deep architectures; see Table 1 for some comparisons.

**Contributions.** Our focus in this paper is the implicit regularization of (standard) gradient descent for high dimensional sparse regression, namely *implicit sparse regularization*. Let us assume a ground-truth sparse linear model and suppose we observe $n$ noisy samples $(\mathbf{x}_i, y_i)$, such that $\mathbf{y} = \mathbf{X}\mathbf{w}^* + \boldsymbol{\xi}$; a more formal setup is given in Section 2. Using the samples, we consider gradient descent on a squared loss $\|\mathbf{X}\mathbf{w} - \mathbf{y}\|^2$ with no explicit sparsity regularization. Instead, we write the parameter vector $\mathbf{w}$ in the form $\mathbf{w} = \mathbf{u}^N - \mathbf{v}^N$ with $N \geq 2$. Now, the regression function $f(\mathbf{x}, \mathbf{u}, \mathbf{v}) = \langle \mathbf{x}, \mathbf{u}^N - \mathbf{v}^N \rangle$ can be viewed as a depth-$N$ *diagonal linear network* [18]. Minimizing the (now non-convex) loss over $\mathbf{u}$ and $\mathbf{v}$ with gradient descent is then analogous to training this depth-$N$ network.

Our main contributions are the following. We characterize the impact of both the depth and early stopping for this non-convex optimization problem. Along the way, we also generalize the results of [11] for $N > 2$. We show that under a general depth parameter $N$ and an incoherence assumption on the design matrix, gradient descent with early stopping achieves minimax optimal recovery with sufficiently small initialization $\mathbf{w}_0$ and step size $\eta$. The choice of step size is of order $O(1/N^2)$. Moreover, the upper bound of the initialization, as well as the early-stopping window, increase with $N$, suggesting that depth leads to a more accessible generalizable solution on gradient trajectories.

**Techniques.** At a high level, our work continues the line of work on implicit bias initiated in [11, 12, 18] and extends it to the deep setting. Table 1 highlights key differences between our work and [11, 13, 18]. Specifically, Woodworth et al. [18] study the *interpolation* given by the gradient flow of the squared-error loss function. Vaskevicius et al. [11] analyze the finite gradient descent and characterize the implicit sparse regularization on the *recovery* of true parameters with $N = 2$. Lastly, Gissin et al. [13] discover the incremental learning dynamic of gradient flow for general $N$ but in an idealistic model setting where $\mathbf{u} \succeq 0$, $\mathbf{v} = 0$, $\boldsymbol{\xi} = 0$, uncorrelated design and with infinitely many samples.

Table 1: Comparisons with closely related recent work. GF/GD: gradient flow/descent, respectively.

| | Design Matrix | In Noise | Depth | Early Stopping | GD vs. GF | Remark |
|---|---|---|---|---|---|---|
| Vaskevicius et al. (2020) [11] | RIP | ✓ | $N = 2$ | ✓ | GD | recovery |
| Gissin et al. (2020) [13] | uncorrelated | ✗ | $N = 2, N > 2$ | ✗ | GF | interpolation |
| Woodworth et al. (2020) [18] | ✗ | ✗ | $N = 2, N > 2$ | ✗ | GF | interpolation |
| This paper | $\mu$-coherence | ✓ | $N > 2$ | ✓ | GD | recovery |

At first glance, one could attempt a straightforward extension of the proof techniques in [11] to general settings of $N > 2$. However, this turns out to be very challenging. Consider even the simplified case where the true model $\mathbf{w}^\star$ is non-negative, the design matrix is unitary (i.e., $n^{-1}\mathbf{X}^\top\mathbf{X} = \mathbf{I}$), and the noise is absent ($\boldsymbol{\xi} = 0$); this is the setting studied in [13]. For each entry $w_i$ of $\mathbf{w}$, the $t^{\text{th}}$ iterate of gradient descent over the depth-$N$ reparametrized model is given by:

$$w_{i,t+1} = w_{i,t} \left( 1 + w_{i,t}^{1-\frac{2}{N}} (w_i^\star - w_{i,t}) \right)^N,$$

which is no longer a simple multiplicative update. As pointed out in [13] (see their Appendix C), the recurrence relation is not analytically solvable due to the presence of the (pesky) term $w_{i,t}^{1-\frac{2}{N}}$ when $N > 2$. Moreover, this extra term $w_{i,t}^{1-\frac{2}{N}}$ leads to widely divergent growth rates of weights with different magnitudes, which further complicates analytical bounds. To resolve this and rigorously analyze the dynamics for $N > 2$, we rely on a novel first order, continuous approximation to study growth rates without requiring additional assumptions on gradient flow, and carefully bound the approximation error due to finite step size; see Section 4.

## 2 Setup

**Sparse regression/recovery.** Let $\mathbf{w}^\star \in \mathbb{R}^p$ be a $p$-dimensional sparse vector with $k$ non-zero entries. Assume that we observe $n$ data points $(\mathbf{x}_i, y_i) \in \mathbb{R}^p \times \mathbb{R}$ such that $y_i = \langle \mathbf{x}_i, \mathbf{w}^\star \rangle + \xi_i$ for $i = 1, \ldots, n$, where $\boldsymbol{\xi} = (\xi_1, \ldots, \xi_n)$ is the noise vector. We do not assume any particular scaling between the number of observations $n$ and the dimension $p$. Due to the sparsity of $\mathbf{w}^\star$, however, we allow $n \ll p$.

The linear model can be expressed in the matrix-vector form:
$$\mathbf{y} = \mathbf{X}\mathbf{w}^\star + \boldsymbol{\xi}, \tag{1}$$
with the $n \times p$ design matrix $\mathbf{X} = [\mathbf{x}_1^\top, \ldots, \mathbf{x}_n^\top]^\top$, where $\mathbf{x}_i$ denotes the $i^{\text{th}}$ *row* of $\mathbf{X}$. We also denote $\mathbf{X} = [\mathbf{X}_1, \ldots, \mathbf{X}_p]$, where $\mathbf{X}_i$ denotes the $i^{\text{th}}$ *column* of $\mathbf{X}$.

The goal of sparse regression is to estimate the unknown, sparse vector $\mathbf{w}^\star$ from the observations. Over the past two decades, this problem has been a topic of active research in statistics and signal processing [25]. A common approach to sparse regression is penalized least squares with sparsity-induced regularization such as $\ell_0$ or $\ell_1$ penalties/constraints, leading to several well-known estimators [25, 26, 27] and algorithms [28, 29]. Multiple estimators enjoy optimal statistical and algorithmic recovery guarantees under some conditions of the design matrix $\mathbf{X}$ (e.g., RIP [30]) and the noise $\boldsymbol{\xi}$.

We deviate from the standard penalized least squares formulation and instead learn $\mathbf{w}^*$ via a polynomial parametrization:
$$\mathbf{w} = \mathbf{u}^N - \mathbf{v}^N, \quad \mathbf{u}, \mathbf{v} \in \mathbb{R}^p,$$
where $N \geq 2$ and $\mathbf{z}^N = [z_1^N, \ldots, z_p^N]^\top$ for any $\mathbf{z} = [z_1, \ldots, z_N]^\top \in \mathbb{R}^p$. The regression function $f(\mathbf{x}, \mathbf{u}, \mathbf{v}) = \langle \mathbf{x}, \mathbf{u}^N - \mathbf{v}^N \rangle$ induced by such a parametrization is equivalent to a $N$-layer diagonal linear network [18] with $2p$ hidden neurons and the diagonal weight matrix shared across all layers.

Given the data $\{\mathbf{X}, \mathbf{y}\}$ observed in (1), we analyze gradient descent with respect to the new parameters $\mathbf{u}$ and $\mathbf{v}$ over the mean squared error loss without explicit regularization:
$$\mathcal{L}(\mathbf{u}, \mathbf{v}) = \frac{1}{n} \left\| \mathbf{X}(\mathbf{u}^N - \mathbf{v}^N) - \mathbf{y} \right\|_2^2, \quad \mathbf{u}, \mathbf{v} \in \mathbb{R}^p.$$

Even though the loss function yields the same value for the two parametrizations, $\mathcal{L}(\mathbf{u}, \mathbf{v})$ is non-convex in $\mathbf{u}$ and $\mathbf{v}$. Unlike several recent studies in implicit regularization for matrix factorization and regression [16, 18, 13], we consider the noisy setting, which is more realistic and leads to more insights into the bias induced during the optimization. Because of noise, the loss evaluated at the ground truth (i.e., any $\mathbf{u}, \mathbf{v}$ such that $\mathbf{w}^\star = \mathbf{u}^N - \mathbf{v}^N$) is not necessarily zero or even minimal.

**Gradient descent.** The standard gradient descent update over $\mathcal{L}(\mathbf{u}, \mathbf{v})$ reads as:
$$\mathbf{u}_0 = \mathbf{v}_0 = \alpha\mathbf{1},$$
$$(\mathbf{u}_{t+1}, \mathbf{v}_{t+1}) = (\mathbf{u}_t, \mathbf{v}_t) - \eta \frac{\partial \mathcal{L}(\mathbf{u}_t, \mathbf{v}_t)}{\partial(\mathbf{u}_t, \mathbf{v}_t)}, \quad t = 0, 1, \ldots. \tag{2}$$

Here, $\eta > 0$ is the step size and $\alpha > 0$ is the initialization of $\mathbf{u}, \mathbf{v}$. In general, we analyze the algorithm presented in (2), and at each step $t$, we can estimate the signal of interest by simply calculating $\mathbf{w}_t = \mathbf{u}_t^N - \mathbf{v}_t^N$. We consider constant initialization for simplicity sake. Our results apply for random initialization concentrating on a small positive region with a probabilistic statement.

Vaskevicius et al. [11] establish the implicit sparse regularization of gradient descent for $N = 2$ and show minimax optimal recovery, provided sufficiently small $\alpha$ and early stopping. Our work aims to generalize that result to $N > 2$ and characterize the role of $N$ in convergence.

**Notation.** We define $S = \{i \in \{1, \ldots, p\} : w_i^\star \neq 0\}$ and $S^c = \{1, \ldots, p\} \backslash S$. The largest and smallest absolute value on the support is denoted as $w_{\max}^\star = \max_{i \in S} |w_i^\star|$ and $w_{\min}^\star = \min_{i \in S} |w_i^\star|$. We use $\mathbf{1}$ to denote the vector of all ones and $\mathbf{1}_S$ denotes the vector whose elements on $S$ are all one and 0 otherwise. Also, $\odot$ denotes coordinate-wise multiplication. We denote $\mathbf{s}_t = \mathbf{1}_S \odot \mathbf{w}_t$ and $\mathbf{e}_t = \mathbf{1}_{S^c} \odot \mathbf{w}_t$ meaning the signal part and error part at each time step $t$. We use $\wedge$ and $\vee$ to denote the pointwise maximum and minimum. The coordinate-wise inequalities are denoted as $\succeq$. We denote inequalities up to multiplicative absolute constants by $\lesssim$, which means that they do not depend on any parameters of the problem.

**Definition 1.** *Let $\mathbf{X} \in \mathbb{R}^{n \times p}$ be a matrix with $\ell_2$-normalized columns $\mathbf{X}_1, \ldots, \mathbf{X}_p$, i.e., $\|\mathbf{X}_i\|_2 = 1$ for all $i$. The coherence $\mu = \mu(\mathbf{X})$ of the matrix $\mathbf{X}$ is defined as*

$$\mu := \max_{1 \leq i \neq j \leq p} |\langle \mathbf{X}_i, \mathbf{X}_j \rangle|.$$

*The matrix $\mathbf{X}$ is said to be satisfying $\mu$-incoherence.*

The coherence is a measure for the suitability of the measurement matrix in compressive sensing [31]. In general, the smaller the coherence, the better the recovery algorithms perform. There are multiple ways to construct a sensing matrix with low-incoherence. One of them is based on the fact that sub-Gaussian matrices satisfy low-incoherence property with high probability [32, 33]. In contrast to the coherence, the Restricted Isometry Property (RIP) is a powerful performance measure for guaranteeing sparse recovery and has been widely used in many contexts. However, verifying the RIP for deterministically constructed design matrices is NP-hard. On the other hand, coherence is a computationally tractable measure and its use in sparse regression is by now classical [33, 34]. Therefore, in contrast with previous results [11] (which assumes RIP), the assumptions made in our main theorems are verifiable in polynomial time.

## 3 Main Results

We now introduce several quantities that are relevant for our main results. First, the condition number $r := w_{\max}^\star / w_{\min}^\star$ plays an important role when we work on the incoherence property of the design matrix. Next, we require an upper bound on the initialization $\alpha$, which depends on the following terms:

$$\Phi(w_{\max}^\star, w_{\min}^\star, \epsilon, N) := \left(\frac{1}{8}\right)^{2/(N-2)} \wedge \left(\frac{(w_{\max}^\star)^{(N-2)/N}}{\log \frac{w_{\max}^\star}{\epsilon}}\right)^{2/(N-2)} \wedge \left(\frac{(w_{\min}^\star)^{(N-2)/N}}{\log \frac{w_{\min}^\star}{\epsilon}}\right)^{4/(N-2)},$$

$$\Psi(w_{\min}^\star, N) := (2 - 2^{\frac{N-2}{N}})^{\frac{1}{N-2}} (w_{\min}^\star)^{\frac{1}{N}} \wedge 2^{\frac{3}{N}} (2^{\frac{1}{N}} - 1)^{\frac{1}{N-2}} (w_{\min}^\star)^{\frac{1}{N}}.$$

Finally, define

$$\zeta := \frac{1}{5} w_{\min}^\star \vee \frac{200}{n} \|\mathbf{X}^\mathsf{T} \boldsymbol{\xi}\|_\infty \vee 200\epsilon.$$

We are now ready to state the main theorem:

**Theorem 1.** *Suppose that $k \geq 1$ and $\mathbf{X}/\sqrt{n}$ satisfies $\mu$-incoherence with $\mu \lesssim 1/kr$. Take any precision $\epsilon > 0$, and let the initialization be such that*

$$0 < \alpha \leq \left(\frac{\epsilon}{p+1}\right)^{4/N} \wedge \Phi(w_{\max}^\star, w_{\min}^\star, \epsilon, N) \wedge \Psi(w_{\min}^\star, N). \tag{3}$$

*For any iteration $t$ that satisfies*

$$T_l(\mathbf{w}^\star, \alpha, N, \eta, \zeta, \epsilon) \leq t \leq T_u(\mathbf{w}^\star, \alpha, N, \eta, \zeta, \epsilon), \tag{4}$$

*where $T_l(\cdot)$ and $T_u(\cdot)$ are given in (25) of the Appendix, the gradient descent algorithm (2) with step size $\eta \leq \frac{\alpha^N}{8N^2 (w_{\max}^\star)^{(3N-2)/N}}$ yields the iterate $\mathbf{w}_t$ with the following property:*

$$|w_{t,i} - w_i^\star| \lesssim \begin{cases} \left\|\frac{1}{n}\mathbf{X}^\mathsf{T}\boldsymbol{\xi}\right\|_\infty \vee \epsilon & \text{if } i \in S \text{ and } w_{\min}^\star \lesssim \left\|\frac{1}{n}\mathbf{X}^\mathsf{T}\boldsymbol{\xi}\right\|_\infty \vee \epsilon, \\ \left|\frac{1}{n}(\mathbf{X}^\mathsf{T}\boldsymbol{\xi})_i\right| \vee k\mu \left\|\frac{1}{n}\mathbf{X}^\mathsf{T}\boldsymbol{\xi} \odot \mathbf{1}_S\right\|_\infty \vee \epsilon & \text{if } i \in S \text{ and } w_{\min}^\star \gtrsim \left\|\frac{1}{n}\mathbf{X}^\mathsf{T}\boldsymbol{\xi}\right\|_\infty \vee \epsilon, \\ \alpha^{N/4} & \text{if } i \notin S. \end{cases} \tag{5}$$

*In the special case $\mathbf{w}^\star = \mathbf{0}$, if $\alpha \leq \left(\frac{\epsilon}{p+1}\right)^{4/N}$, $\eta \leq \frac{1}{N(N-1)\zeta\alpha^{(N-2)/2}}$ and $t \leq T_u(\mathbf{w}^\star, \alpha, N, \eta, \zeta, \epsilon)$, then we have $|w_{t,i} - w_i^\star| \leq \alpha^{N/4}, \forall i$.*

Theorem 1 states the convergence of the gradient descent algorithm (2) in $\ell_\infty$-norm. The exact formula of $T_l(\cdot)$ and $T_u(\cdot)$ is omitted here due to the space limitation. We ensure that $T_u(\cdot) > T_l(\cdot)$ so that there indeed exists some epochs to early stop at. The error bound on the signal is invariant to the choice of $N \geq 2$, and the overall bound generalizes that of [11] for $N = 2$. We also establish the convergence result in $\ell_2$-norm in the following corollary:

**Corollary 1.** *Suppose the noise vector $\boldsymbol{\xi}$ has independent $\sigma^2$-sub-Gaussian entries and $\epsilon = 2\sqrt{\frac{\sigma^2 \log(2p)}{n}}$. Under the assumptions of Theorem 1, the gradient descent algorithm (2) would produce iterate $\mathbf{w}_t$ satisfying $\|\mathbf{w}_t - \mathbf{w}^\star\|_2^2 \lesssim (k\sigma^2 \log p)/n$ with probability at least $1 - 1/(8p^3)$.*

Note that the error bound we obtain is minimax-optimal, which is the same as [11] in the $N = 2$ case. However, with some calculation, the sample complexity we obtain here is $n \gtrsim k^2 r^2$, while the sample complexity in [11] is $n \gtrsim k^2 \log^2 r \log p/k$. Although neither our work nor [11] achieved the optimal sample complexity $k \log p/k$, the goal of this work is to understand how the depth parameter $N$ affects implicit sparse regularization.

Let us now discuss the implications of Theorem 1 and the role of initialization and early stopping:

**(a) Requirement on initialization.** To roughly understand the role of initialization and the effect of $N$, we look at the non-negative case where $\mathbf{w}^\star \succcurlyeq 0$ and $\mathbf{w} = \mathbf{u}^N$. This simplifies our discussion while still capturing the essential insight of the general setting. At each step, the "update" on $\mathbf{w}$ can be translated from the corresponding gradient update of $\mathbf{u} = \mathbf{w}^{1/N}$ as

$$\mathbf{w}_0 = \alpha^N \mathbf{1},$$

$$\mathbf{w}_{t+1} = \mathbf{w}_t \odot \left( \mathbf{1} - \frac{2N\eta}{n} \left( \mathbf{X}^\mathsf{T} \mathbf{X}(\mathbf{w}_t - \mathbf{w}^*) - \mathbf{X}^\mathsf{T} \boldsymbol{\xi} \right) \odot \mathbf{w}_t^{(N-2)/N} \right)^N. \quad (6)$$

In order to guarantee the convergence, we require the initialization $\alpha$ to be sufficiently small so the error outside the support can be controlled. On the other hand, too small initialization slows down the convergence of the signal. Interestingly, the choice of $N$ affects the allowable initialization $\alpha$ that results in guarantees on the entries inside and outside the support.

Specifically, the role of $N$ is played by the term $\mathbf{w}_t^{(N-2)/N}$ in (6), which simply disappears as $N = 2$. Since this term only affects the update of $\mathbf{w}_{t+1}$ entry-wise, we only look at a particular entry $w_t$ of $\mathbf{w}_t$. Let $w_t$ represent an entry outside the support. For $N > 2$, the term $w_t^{(N-2)/N}$ is increasingly small as $N$ increases and $w_t < 1$. Therefore, with a small initialization, it remains true that $w_t < 1$ for the early iterations. Intuitively, this suggests that the requirement on the upper bound of the initialization would become looser when $N$ gets larger. This indeed aligns with the behavior of the upper bound we derive in our theoretical results. Since $\alpha = w_0^{1/N}$ increases naturally with $N$, we fix $w_0 = \alpha^N$ instead of $\alpha$ to mimic the same initialization in terms of $w_0$, for the following comparison.

We formalize this insight in Theorem 3 in Appendix A and show the convergence of (6) under the special, non-negative case. Note that, in terms of initialization requirement, the only difference from Theorem 1 is that we no longer require the term $\Psi(w_{\min}^\star, N)$ in (3).

**Remark 1.** *We investigate how the depth $N$ influences the requirement on initialization due to the change on gradient dynamics. We rewrite $\Phi(w_{\max}^\star, w_{\min}^\star, \epsilon, N)$ in terms of $w_0 = \alpha^N$, and therefore the upper bound for $w_0$ under the simplified setting of non-negative signals (Theorem 3) is*

$$w_0 \leq \left( \frac{1}{8} \right)^{2N/(N-2)} \wedge \left( \frac{(w_{\max}^\star)^{(N-2)/N}}{\log \frac{w_{\max}^\star}{\epsilon}} \right)^{2N/(N-2)} \wedge \left( \frac{(w_{\min}^\star)^{(N-2)/N}}{\log \frac{w_{\min}^\star}{\epsilon}} \right)^{4N/(N-2)}.$$

*We start by analyzing each term in the upper bound. First, we notice that $\left( \frac{1}{8} \right)^{2N/(N-2)}$ is increasing with respect to $N$. For the second term,*

$$\left( \frac{(w_{\max}^\star)^{(N-2)/N}}{\log \frac{w_{\max}^\star}{\epsilon}} \right)^{2N/(N-2)} = \frac{(w_{\max}^\star)^2}{(\log \frac{w_{\max}^\star}{\epsilon})^{2N/(N-2)}},$$

*the denominator gets smaller as $N$ increases when we pick the error tolerance parameter $\epsilon$ small. Therefore, we get that the second term is getting larger as $N$ increases. The last term*

$\left( \frac{(w_{\min}^{\star})^{(N-2)/N}}{\log \frac{w_{\min}^{\star}}{\epsilon}} \right)^{4N/(N-2)}$ *follows a similar argument. We see that it is possible to pick a larger initialization $w_0 = \alpha^N$ for larger $N$. We will demonstrate that below in our experiments.*

**(b) Early stopping.** Early stopping is shown to be crucial, if not necessary, for implicit sparse regularization [11, 12]. Interestingly, [13, 18] studied the similar depth-$N$ polynomial parametrization but did not realize the need of early stopping due to an oversimplification in the model. We will discuss this in details in Section 4.1. We are able to explicitly characterize the window of the number of iterations that are sufficient to guarantee the optimal result. In particular, we get a lower bound of the window size for early stopping to get a sense of how it changes with different $N$.

**Theorem 2** (Informal). *Define the early stopping window size as $T_u(\mathbf{w}^{\star}, \alpha, N, \eta, \zeta, \epsilon) - T_l(\mathbf{w}^{\star}, \alpha, N, \eta, \zeta, \epsilon)$, the difference between the upper bound and lower bound of the number of iterations in (4) of Theorem 1. Fixing $\alpha$ and $\eta$ for all $N$, the early stopping window size is increasing with $N$ under mild conditions.*

We defer the formal argument and proof of Theorem 2 to Appendix D.3. We note that the window we obtain in Theorem 1 is not necessarily the largest window that allows the guarantee, and hence the early stopping window size can be effectively regarded a lower bound of that derived from the largest window. We note that a precise characterization of the largest window is difficult. Although we only show that this lower bound increases with $N$, we see that the conclusion matches empirically with the largest window. Fix the same initialization $\alpha = 0.005$ and step size $\eta = 0.01$ for $N = 2, 3, 4$, we show the coordinate path in Figure 1. We can see that as $N$ increases, the early stopping window increases and the error bound captures the time point that needs stopping quite accurately. The experimental details and more experiments about early stopping is presented in Section 5.

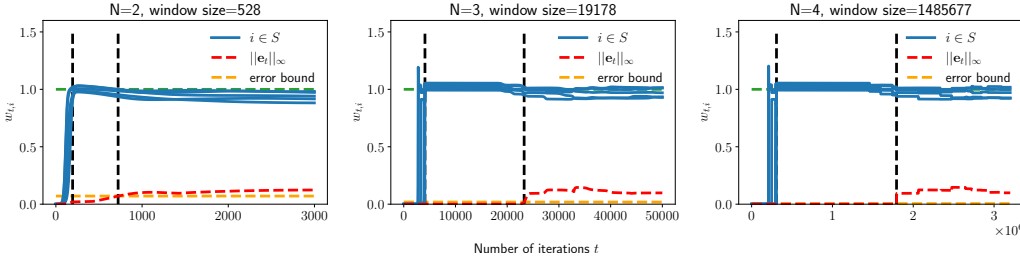

Figure 1: The black line indicates the early stopping window for different $N = 2, 3, 4$. The blue line is the coordinate path for each entry on the support. The red line indicates the absolute value of the largest entry on the coordinate path outside the support. We use the orange line to indicate the requirement outside the support for early stopping.

**Remark 2.** *Similar to Theorem 2, we look at how initialization scale affects the early stopping window for any fixed $N > 2$. With $\eta$ fixed, the early stopping window is increasing as the initialization $\alpha$ decreases.*

We defer the detailed calculation to Section D.4. This generalizes the finding that vanishing initialization increases the gap between the phase transition times in [14] from $N = 2$ to any $N > 2$.

## 4  Proof Ingredients

The goal of this paper is to understand how generalization and gradient dynamics change with different $N > 2$. For $N = 2$, gradient descent yields both statistically and computationally optimal recovery under the RIP assumption [11]. The matrix formulation of the same type of parametrization is considered in the setting of low-rank matrix recovery, and exact recovery can be achieved in the noiseless setting [15, 16]. The key proof ingredient is to reduce the convergence analysis to one-dimensional iterates and differentiate the convergence on the support from the error outside the support. Before we get into that, we conduct a simplified gradient flow analysis.

### 4.1 A Simplified Analysis

Consider a simplified problem where the target signal $\mathbf{w}^\star$ is non-negative, $n^{-1}\mathbf{X}^\mathsf{T}\mathbf{X} = \mathbf{I}$ and the noise is absent. We omit the reparametrization of $\mathbf{v}^N$ like before and the gradient descent updates on $\mathbf{u}$ will be independent for each coordinate. The gradient flow dynamics of $\mathbf{w} = \mathbf{u}^N$ is derived as

$$\frac{\partial w_i}{\partial t} = \frac{\partial w_i}{\partial u_i}\frac{\partial u_i}{\partial t} = -\frac{\partial w_i}{\partial u_i}\frac{\partial \mathcal{L}}{\partial u_i} = 2N^2(w_i^\star - w_i)w_i^{2-\frac{2}{N}}, \tag{7}$$

for all $i \in \{1, 2, \ldots, p\}$. Notice that $w_i$ increases monotonically and converges to $w_i^\star$ if $w_i^\star$ is positive or otherwise keeps decreasing and converges to 0 if $w_i^\star = 0$. As such, we can easily distinguish the support and non-support. In fact, gradient flow with dynamics as in (7) would exhibit a behavior of "incremental learning" — the entries are learned separately, one at a time [13]. However, with the presence of noise and perturbation arising from correlated designs, the gradient flow may end up over-fitting the noise. Therefore, early stopping as well as the choice of step size is crucial for obtaining the desired solution [11]. We use (7) to obtain a gradient descent update:

$$w_{i,t+1} = w_{i,t}(1 + 2N^2\eta(w_i^\star - w_{i,t})w_{i,t}^{1-\frac{2}{N}}). \tag{8}$$

The gradient descent with $N = 2$ is analyzed in [11]. However, when $N > 2$, the presence of $w_{i,t}^{1-\frac{2}{N}}$ imposes an asymmetrical effect on the gradient dynamics. The difficulty of analyzing such gradient descent (8) is pointed out in [13]. More specifically, the recurrence relation is not solvable. However, gradient descent updates still share similar dynamics with the idealized gradient flow in (7). Inspired by this effect, we are able to show that the entries inside the support and those outside the support are learned separately with a practical optimization algorithm shown in (2) and (12). As a result, we are able to explore how the depth $N$ affects the choice of step size and early stopping criterion.

### 4.2 Proof Sketch

**Growth rate of gradient descent.** We adopt the same decomposition as illustrated in [11], and define the following error sequences:

$$\mathbf{b}_t = \frac{1}{n}\mathbf{X}^\mathsf{T}\mathbf{X}\mathbf{e}_t - \frac{1}{n}\mathbf{X}^\mathsf{T}\boldsymbol{\xi}, \quad \mathbf{p}_t = \left(\frac{1}{n}\mathbf{X}^\mathsf{T}\mathbf{X} - \mathbf{I}\right)(\mathbf{s}_t - \mathbf{w}^\star), \tag{9}$$

where $\mathbf{e}_t$ and $\mathbf{s}_t$ stand for error and signal accordingly, and the definitions can be found in (13) in Appendix. We can then write the updates on $\mathbf{s}_t$ and $\mathbf{e}_t$ as

$$\begin{aligned}
\mathbf{s}_{t+1} &= \mathbf{s}_t \odot (\mathbf{1} - 2N\eta(\mathbf{s}_t - \mathbf{w}^\star + \mathbf{p}_t + \mathbf{b}_t) \odot \mathbf{s}_t^{(N-2)/N})^N, \\
\mathbf{e}_{t+1} &= \mathbf{e}_t \odot (\mathbf{1} - 2N\eta(\mathbf{p}_t + \mathbf{b}_t) \odot \mathbf{e}_t^{(N-2)/N})^N.
\end{aligned} \tag{10}$$

To illustrate the idea, we think of the one-dimensional updates $\{s_t\}_{t\geq 0}$ and $\{e_t\}_{t\geq 0}$, ignore the error perturbations $\mathbf{p}_t$ and $\mathbf{b}_t$ in the signal updates $\{s_t\}_{t\geq 0}$, and treat $\|\mathbf{p}_t + \mathbf{b}_t\|_\infty \leq B$ in the error updates $\{e_t\}_{t\geq 0}$.

$$s_{t+1} = s_t(1 - 2N\eta(s_t - w^\star)s_t^{(N-2)/N})^N, \quad e_{t+1} = e_t(1 - 2N\eta Be_t^{(N-2)/N})^N. \tag{11}$$

We use the continuous approximation to study the discrete updates. Therefore, we can borrow many insights from the analysis about gradient flow to overcome the difficulties caused by $w_{i,t}^{1-\frac{2}{N}}$ as pointed out in equation (8). With a proper choice of step size $\eta$, the number of iterations $T_l$ for $s_t$ converging to $w^\star$ is derived as

$$T_l \leq \sum_{t=0}^{T_l-1} \frac{s_{t+1} - s_t}{2N^2\eta(w^\star - s_t)s_t^{(2N-2)/N}} \quad \leq \frac{1}{N^2\eta w^\star}\int_{\alpha^N}^{w^\star}\frac{1}{s^{(2N-2)/N}}ds + \mathcal{O}\left(\frac{w^\star - \alpha^N}{\alpha^{2N-2}}\right).$$

The number of iterations $T_u$ for $e_t$ staying below some threshold $\alpha^{N/4}$ is derived as

$$T_u \geq \sum_{t=0}^{T_u-1}\frac{e_{t+1} - e_t}{4N^2\eta Be_t^{(2N-2)/N}} \geq \frac{1}{4N^2\eta B}\int_{\alpha^N}^{\alpha^{N/4}}\frac{1}{e^{(2N-2)/N}}de.$$

With our choice of coherence $\mu$ in Theorem 1, we are able to control $B$ to be small so that $T_l$ is smaller than $T_u$. This means the entries on the support converge to the true signal while the entries outside the support stay around 0, and we are able to distinguish signals and errors.

**Dealing with negative targets.** We now illustrate the idea about how to generalize the result about non-negative signals to general signals. The exact gradient descent updates on $\mathbf{u}$ and $\mathbf{v}$ are given by:

$$\mathbf{u}_{t+1} = \mathbf{u}_t \odot \left( \mathbf{1} - 2N\eta \left( \frac{1}{n}\mathbf{X}^{\mathsf{T}}(\mathbf{X}(\mathbf{w}_t - \mathbf{w}^\star) - \boldsymbol{\xi}) \odot \mathbf{u}_t^{N-2} \right) \right),$$

$$\mathbf{v}_{t+1} = \mathbf{v}_t \odot \left( \mathbf{1} + 2N\eta \left( \frac{1}{n}\mathbf{X}^{\mathsf{T}}(\mathbf{X}(\mathbf{w}_t - \mathbf{w}^\star) - \boldsymbol{\xi}) \odot \mathbf{v}_t^{N-2} \right) \right). \tag{12}$$

The basic idea is to show that when $w_i^\star$ is positive, $v_i^\star$ remains small up to the early stopping criterion, and when $w_i^\star$ is negative, $u_i^\star$ remains small up to the early stopping criterion. We turn to studying the gradient flow of such dynamics. Write $\mathbf{r}(t) = \frac{1}{n}\mathbf{X}^{\mathsf{T}}(\mathbf{X}(\mathbf{w}(t) - \mathbf{w}^\star) - \boldsymbol{\xi})$. It is easy to verify that the gradient flow has a solution:

$$\mathbf{u}(t) = \left( \alpha^{2-N}\mathbf{1} + 2N(N-2)\eta \int_0^t \mathbf{r}(v)dv \right)^{\frac{1}{2-N}},$$

$$\mathbf{v}(t) = \left( \alpha^{2-N}\mathbf{1} - 2N(N-2)\eta \int_0^t \mathbf{r}(v)dv \right)^{\frac{1}{2-N}}.$$

We may observe some symmetry here, when $u_{i,t}$ is large, $v_{i,t}$ must be small. For the case $w_i > 0$, to ensure the increasing of $u_{i,t}$ and decreasing of $v_{i,t}$ as we desire, the initialization needs to be smaller than $w_i$, which leads to the extra constraint on initialization $\Psi(w_{\min}^\star, \epsilon)$ with order of $\mathcal{O}(w_{\min}^\star)$ as defined before. It remains to build the connection between gradient flow and gradient descent, where again we uses the continuous approximation as before. The detailed derivation is presented in Appendix B.3.

## 5  Simulation Study

We conduct a series of simulation experiments[1] to further illuminate our theoretical findings. Our simulation setup is described as follows. The entries of $\mathbf{X}$ are sampled as i.i.d. Rademacher random variables and the entries of the noise vector $\boldsymbol{\xi}$ are i.i.d. $N(0, \sigma^2)$ random variables. We let $\mathbf{w}^\star = \gamma \mathbf{1}_S$. The values for the simulation parameters are: $n = 500$, $p = 3000$, $k = 5$, $\gamma = 1$, $\sigma = 0.5$ unless otherwise specified. For $\ell_2$-plots each simulation is repeated 30 times, and the median $\ell_2$ error is depicted. The shaded area indicates the region between $25^{\text{th}}$ and $75^{\text{th}}$ percentiles pointwisely.

**Convergence results.** We start by showing that the general choice of $N$ leads to the sparse recovery, similar to $N = 2$ in [11], as shown in our main theorem. We choose different values of $N$ to illustrate the convergence of the algorithm. The result on simulated data is shown in Figure 2, and we defer the result on MNIST to Appendix E. Note that the ranges in the $x$-axes of these figures differ due to different choice of $N$ and $\eta$. We observe that as $N$ increases, the number of iterations increases significantly. This is due to the term $\mathbf{u}^{N-2}$ and $\mathbf{v}^{N-2}$ in (12), and the step size $\eta \approx \frac{1}{N^2}$. With a very small initialization, it takes a large number of iterations to escape from the small region (close to 0).

**Larger initialization.** As discussed in Remark 1, the upper bound on initialization gets larger with larger $N$. We intentionally pick a relatively large $\alpha^N = 2 \times 10^{-3}$ where the algorithm fails to converge for $N = 2$. With the same initialization, the recovery manifests as $N$ increases (Figure 3).

**Early stopping window size.** Apart from the coordinate path shown in Figure 1, we obtain multiple runs and plot the $\log$-$\ell_2$ error (the logarithm of the $\ell_2$-error) of the recovered signals to further confirm the increase of early stopping window, as shown in Section 3. Note that for both Figures 1 and 4, we set $n = 100$ and $p = 200$. Since $\alpha^N$ would decrease quickly with $N$, which would cause the algorithm takes a large number of iterations to escape from the small region. We fix $\alpha^N = 10^{-5}$ instead of fixing $\alpha$ for Figure 4.

**Incremental learning dynamics.** The dynamics of incremental learning for different $N$ is discussed in [13]. The distinct phases of learning are also observed in sparse recovery (Figure 5), though we do

---

[1]The code is available on `https://github.com/jiangyuan2li/Implicit-Sparse-Regularization`.

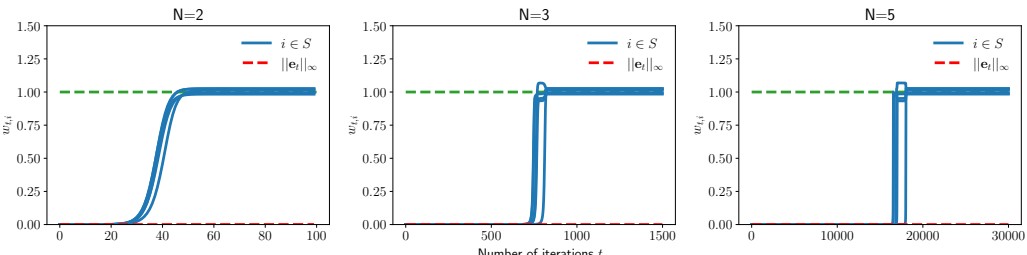

Figure 2: Coordinates paths for different choice of $N = 2, 3, 5$ with $\alpha^N = 10^{-6}$ and $\eta = 1/(5N^2)$.

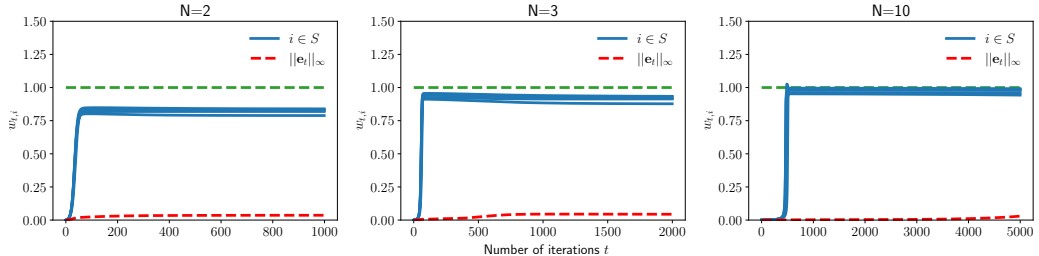

Figure 3: The effect of $N$ on the initialization $\alpha^N$ with $\eta = 1/(5N^2)$.

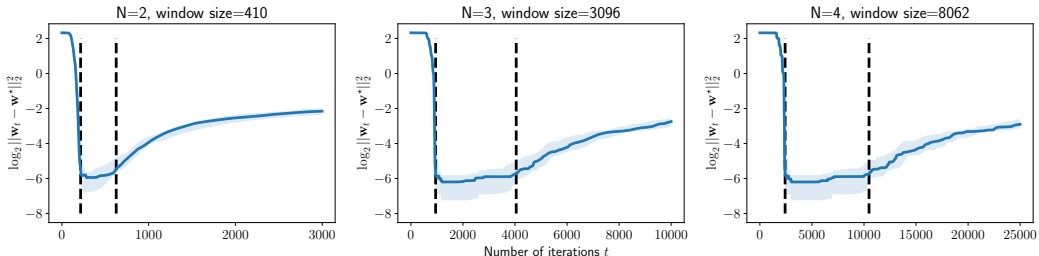

Figure 4: $\log$-$\ell_2$ error of $N = 2, 3, 4$ with the fixed step size $\eta = 0.01$.

not provide a theoretical justification. Larger values of $N$ would lead to more distinct learning phases for entries with different magnitudes under the same initialization $\alpha^N$ and step size $\eta$.

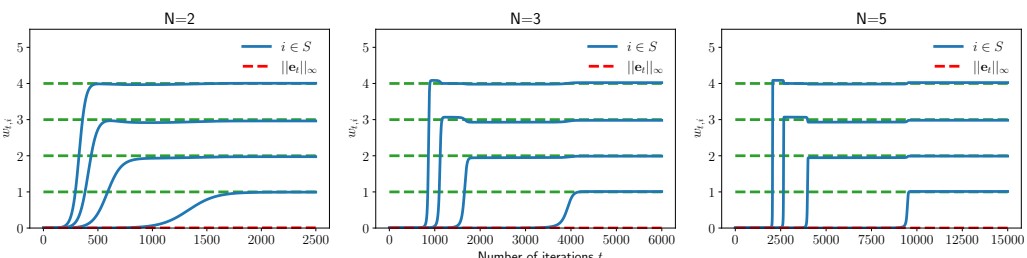

Figure 5: Coordinates paths for $N = 2, 3, 5$. The entries of $\mathbf{w}^\star$ on the support $S$ are now $[1, 2, 3, 4]$. The initialization is $\alpha^N = 10^{-4}$ and the step size is $\eta = 10^{-3}$ for all $N$.

**Kernel regime.** As pointed out in [18], the scale of initialization determines whether the gradient dynamics obey the "kernel" or "rich" regimes for diagonal linear networks. We have carefully analyzed and demonstrated the sparse recovery problem with small initialization, which corresponds to the "rich" regime. To explore the "kernel" regime in a more practical setting, we set $n = 500$, $p = 100$, and the entries of $\mathbf{w}^\star$ are i.i.d. $\mathcal{N}(0, 1)$ random variables. The noise level is $\sigma = 25$, and the initialization and step size is set as $\alpha^N = 1000$ and $\eta = 10^{-7}$ for all $N$. Note that we are not working

in the case $n \ll p$ as [18]. We still observe that the gradient dynamics with large initialization (Figure 6) can be connected to ridge regression if early stopping is deployed.

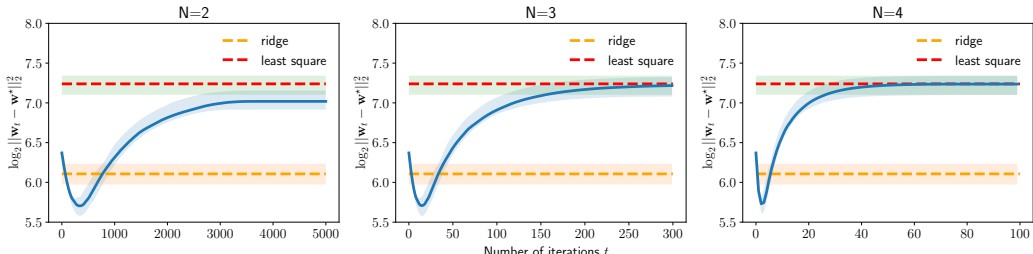

Figure 6: $\log$-$\ell_2$ error of $N = 2, 3, 4$ for a ridge regression setting. The ridge regression solution is selected by 5-fold cross validation.

## 6 Conclusions and Future Work

In this paper, we extend the implicit regularization results in [11] from $N = 2$ to general $N > 2$, and further study how gradient dynamics and early stopping is affected by different choice $N$. We show that the error bound is invariant with different choice of $N$ and yields the minimax optimal rate. The step size is of order $\mathcal{O}(1/N^2)$. The initialization and early stopping window gets larger when increasing $N$ due to the changes on gradient dynamics. The incremental learning dynamics and kernel regime of such parametrizations are empirically shown, however not theoretically justified, which is left for future work.

The convergence result can be further improved by relaxing the requirement on the incoherence of design matrix from $\mu \lesssim \frac{1}{k w^\star_{\max}/w^\star_{\min}}$ to $\mu \lesssim \frac{1}{k \log(w^\star_{\max}/w^\star_{\min})}$, similar to [11]. Overall, we believe that such an analysis and associated techniques could be applied for studying other, deeper nonlinear models in more practical settings.

## Acknowledgements

This work was supported in part by the National Science Foundation under grants CCF-1934904, CCF-1815101, CCF-2005804, and DMS-1711952.

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
