# Appendix

The appendix is organized as follows.

In Appendix A, we present a simplied theorem about non-negative signals and illustrate the idea behind the proof.

In Appendix B, we study the multiplicative updates and build connections to its continuous approximation, which will be used next.

In Appendix C, we provide the proof of propositions and technical lemmas in Appendix A.

In Appendix D, we prove the main results stated in the paper.

In Appendix E, we provide the experimental results on real-world datasets to illustrate the effectiveness of the proposed algorithm.

## A    Proof for Non-negative Signals

We mainly follow the proof structure from [11] to obtain the convergence of similar gradient descent algorithm for the case $N = 2$, which is a limiting case of ours. We will demonstrate how gradient dynamics changes with $N > 2$, which requires us to study the growth rate of error and convergence rate more carefully.

In this section, we will start with the general set up and provide a simplified version of Theorem 1 about non-negative signals.

### A.1    Setup

The gradients of $\mathcal{L}(\mathbf{u}, \mathbf{v})$ with respect to $\mathbf{u}, \mathbf{v}$ read as

$$\nabla_{\mathbf{u}}\mathcal{L}(\mathbf{w}) = \frac{2N}{n}\mathbf{X}^{\mathsf{T}}(\mathbf{X}\mathbf{w} - \mathbf{y}) \odot \mathbf{u}^{N-1}$$
$$\nabla_{\mathbf{v}}\mathcal{L}(\mathbf{w}) = -\frac{2N}{n}\mathbf{X}^{\mathsf{T}}(\mathbf{X}\mathbf{w} - \mathbf{y}) \odot \mathbf{v}^{N-1}.$$

With the step size $\eta$, the gradient descent updates on $\mathbf{u}_t$ and $\mathbf{v}_t$ simply are

$$\mathbf{u}_{t+1} = \mathbf{u}_t \odot \left(\mathbf{1} - 2N\eta \left(\frac{1}{n}\mathbf{X}^{\mathsf{T}}(\mathbf{X}(\mathbf{w}_t - \mathbf{w}^{\star}) - \boldsymbol{\xi}) \odot \mathbf{u}_t^{N-2}\right)\right),$$
$$\mathbf{v}_{t+1} = \mathbf{v}_t \odot \left(\mathbf{1} + 2N\eta \left(\frac{1}{n}\mathbf{X}^{\mathsf{T}}(\mathbf{X}(\mathbf{w}_t - \mathbf{w}^{\star}) - \boldsymbol{\xi}) \odot \mathbf{v}_t^{N-2}\right)\right).$$

Let $\mathbf{w}_t = \mathbf{w}_t^+ - \mathbf{w}_t^-$ where $\mathbf{w}_t^+ := \mathbf{u}_t^N$ and $\mathbf{w}_t^- := \mathbf{v}_t^N$ with the power taken element-wisely. We denote $S$ as the support of $\mathbf{w}^{\star}$, and let $S^+ = \{i|w_i^{\star} > 0\}$ denote the index set of coordinates with positive values, and $S^- = \{i|w_i^{\star} < 0\}$ denote the index set of coordinates with negative values. Therefore $S = S^+ \cup S^-$ and $S^+ \cap S^- = \emptyset$. Then define the following signal and noise-related quantities:

$$\begin{aligned}
\mathbf{s}_t &:= \mathbf{1}_{S^+} \odot \mathbf{w}_t^+ - \mathbf{1}_{S^-} \odot \mathbf{w}_t^-, \\
\mathbf{e}_t &:= \mathbf{1}_{S^c} \odot \mathbf{w}_t + \mathbf{1}_{S^-} \odot \mathbf{w}_t^+ - \mathbf{1}_{S^+} \odot \mathbf{w}_t^-, \\
\mathbf{b}_t &:= \frac{1}{n}\mathbf{X}^{\mathsf{T}}\mathbf{X}\mathbf{e}_t - \frac{1}{n}\mathbf{X}^{\mathsf{T}}\boldsymbol{\xi}, \\
\mathbf{p}_t &:= \left(\frac{1}{n}\mathbf{X}^{\mathsf{T}}\mathbf{X} - \mathbf{I}\right)(\mathbf{s}_t - \mathbf{w}^{\star}).
\end{aligned} \tag{13}$$

Let $\alpha^N$ be the initial value for each entry of $\mathbf{w}$ and rewrite the updates on $\mathbf{w}_t$, $\mathbf{w}_t^+$ and $\mathbf{w}_t^-$ in a more succinct way:

$$
\begin{aligned}
\mathbf{w}_0^+ &= \mathbf{w}_0^- = \alpha^N, \\
\mathbf{w}_t &= \mathbf{w}_t^+ - \mathbf{w}_t^-, \\
\mathbf{w}_{t+1}^+ &= \mathbf{w}_t^+ \odot \left( 1 - 2N\eta \left( \mathbf{s}_t - \mathbf{w}^\star + \mathbf{p}_t + \mathbf{b}_t \right) \odot (\mathbf{w}_t^+)^{(N-2)/N} \right)^N, \\
\mathbf{w}_{t+1}^- &= \mathbf{w}_t^- \odot \left( 1 + 2N\eta \left( \mathbf{s}_t - \mathbf{w}^\star + \mathbf{p}_t + \mathbf{b}_t \right) \odot (\mathbf{w}_t^-)^{(N-2)/N} \right)^N.
\end{aligned}
\tag{14}
$$

When our target $\mathbf{w}^\star$ is with non-negative entries, the design of $\mathbf{v}_t$ is no longer needed and the algorithm could be simplied to the following form.

$$
\begin{aligned}
\mathbf{w}_0^+ &= \mathbf{u}_0^N = \alpha^N, \\
\mathbf{w}_t^+ &= \mathbf{u}_t^N, \\
\mathbf{w}_{t+1}^+ &= \mathbf{w}_t^+ \odot \left( 1 - 2N\eta \left( \mathbf{s}_t - \mathbf{w}^\star + \mathbf{p}_t + \mathbf{b}_t \right) \odot (\mathbf{w}_t^+)^{(N-2)/N} \right)^N
\end{aligned}
\tag{15}
$$

The results in this section are all about updates in equation (15), and will be generalized to updates in equation (14) in Section D.

## A.2 The Key Propositions

Starting from $t = 0$, we have $\|\mathbf{s}_0 - \mathbf{w}^\star\|_\infty \lesssim \mathcal{O}(w_{\max}^\star)$ and $\|\mathbf{e}_0\|_\infty \leq \alpha^N$. The idea of proposition 1 is to show that after some certain number of iterations $t$, we obtain $\|\mathbf{s}_t - \mathbf{w}^\star\|_\infty \lesssim \mathcal{O}(w_{\min}^\star)$ and $\|\mathbf{e}_t\|_\infty \leq \alpha^{N/2}$. Proposition 2 further reduces the approximation error from $\mathcal{O}(w_{\min}^\star)$ to $\mathcal{O}(\|\frac{1}{n}\mathbf{X}^\mathsf{T}\boldsymbol{\xi}\|_\infty)$ if possible, while still maintaining $\|\mathbf{e}_t\|_\infty \leq \alpha^{N/4}$.

**Proposition 1.** *Consider the updates in equations* (15). *Fix any* $0 < \zeta \leq w_{\max}^\star$ *and let* $\gamma = C_\gamma \frac{w_{\min}^\star}{w_{\max}^\star}$ *where $C_\gamma$ is some small enough absolute constant. Suppose the error sequences $(\mathbf{b}_t)_{t\geq 0}$ and $(\mathbf{p}_t)_{t\geq 0}$ for any $t \geq 0$ satisfy the following:*

$$
\begin{aligned}
\|\mathbf{b}_t\|_\infty &\leq C_b \zeta - \alpha^{N/4}, \\
\|\mathbf{p}_t\|_\infty &\leq \gamma \|\mathbf{s}_t - \mathbf{w}^\star\|_\infty,
\end{aligned}
$$

*where $C_b$ is some small enough absolute constants. If the initialization satisfies*

$$
\alpha \leq \left( \frac{1}{8} \right)^{2/(N-2)} \wedge \left( \frac{(w_{\max}^\star)^{(N-2)/N}}{\log \frac{w_{\max}^\star}{\epsilon}} \right)^{2/(N-2)},
$$

*and the step size $\eta \leq \frac{\alpha^N}{8N^2\zeta^{(3N-2)/N}}$, then for any $T_1 \leq T \leq T_2$ where*

$$
\begin{aligned}
T_1 &= \frac{75}{16\eta N^2 \zeta^{(2N-2)/N}} \log \frac{|w_{\max}^\star - \alpha^N|}{\epsilon} + \frac{15}{8N(N-2)\eta\zeta\alpha^{(N-2)}}, \\
T_2 &= \frac{5}{N(N-1)\eta\zeta} \left( \frac{1}{\alpha^{(N-2)}} - \frac{1}{\alpha^{(N-2)/2}} \right),
\end{aligned}
$$

*and any $0 \leq t \leq T$, we have*

$$
\begin{aligned}
\|\mathbf{s}_T - \mathbf{w}^\star\|_\infty &\leq \zeta, \\
\|\mathbf{e}_t\|_\infty &\leq \alpha^{N/2}.
\end{aligned}
$$

Note that the requirement on $\|\mathbf{b}_t\|_\infty \leq C_b\zeta - \alpha^{N/4}$ can be relaxed to $\|\mathbf{b}_t\|_\infty \leq C_b\zeta$ when we just consider the updates in equation (15). However, we still consider the stronger requirement in order to further generalize to updates in equation (14) later.

**Proposition 2.** *Consider the updates in equations* (15). *Fix any* $0 < \zeta \leq w^\star_{\max}$ *and suppose that the error sequences* $(\mathbf{b}_t)_{t\geq 0}$ *and* $(\mathbf{p}_t)_{t\geq 0}$ *for any* $t \geq 0$ *satisfy*

$$B = \|\mathbf{b}_t\|_\infty + \|\mathbf{p}_t\|_\infty \leq \frac{1}{200}w^\star_{\min}$$

$$\|\mathbf{b}_t \odot \mathbf{1}_i\|_\infty \leq B_i \leq \frac{1}{10}w^\star_{\min},$$

$$\|\mathbf{p}_t\|_\infty \leq \frac{1}{20}\|\mathbf{s}_0 - \mathbf{w}^\star\|_\infty.$$

*Suppose that*

$$\alpha \leq \left(\frac{1}{4}\right)^{2/(N-2)} \wedge \left(\frac{(w^\star_{\min})^{(N-2)/N}}{\log\frac{w^\star_{\min}}{\epsilon}}\right)^{4/(N-2)},$$

$$\|\mathbf{s}_0 - \mathbf{w}^\star\|_\infty \leq \frac{1}{5}w^\star_{\min},$$

$$\|\mathbf{e}_0\| \leq \alpha^{N/2}.$$

*Let the step size satisfy* $\eta \leq \frac{\alpha^N}{8N^2(w^\star_{\min})^{(3N-2)/N}}$. *Then for any* $T_3 \leq t \leq T_4$,

$$T_3 = \frac{6}{\eta N^2(w^\star_{\min})^{(2N-2)/N}}\log\frac{w^\star_{\min}}{\epsilon},$$

$$T_4 = \frac{25}{N(N-1)\eta w^\star_{\min}}\left(\frac{1}{\alpha^{(N-2)/2}} - \frac{1}{\alpha^{(N-2)/4}}\right),$$

*and any* $i \in S$ *we have*

$$|s_{i,t} - w^\star_i| \lesssim k\mu \max_{j\in S} B_j \vee B_i \vee \epsilon,$$

$$\|\mathbf{e}_t\|_\infty \leq \alpha^{N/4}.$$

### A.3 Technical Lemmas

There are several lemmas, which are about the coherence of the design matrices and the upper bound of subGaussian noise term.

**Lemma 1.** *Suppose that* $\frac{1}{\sqrt{n}}\mathbf{X}$ *is a* $n \times p$ *matrix with* $\ell_2$-*normalized columns and satisfies* $\mu$-*coherence with* $0 \leq \mu \leq 1$. *Then for any vector* $\mathbf{z} \in \mathbb{R}^p$ *we have*

$$\left\|\frac{1}{n}\mathbf{X}^\mathsf{T}\mathbf{X}\mathbf{z}\right\|_\infty \leq p\|\mathbf{z}\|_\infty.$$

**Lemma 2.** *Suppose that* $\frac{1}{\sqrt{n}}\mathbf{X}$ *is a* $n \times p$ $\ell_2$-*normalized matrix satisfying* $\mu$-*incoherence; that is* $\frac{1}{n}|\mathbf{X}_i^\mathsf{T}\mathbf{X}_j| \leq \mu, i \neq j$. *For* $k$-*sparse vector* $\mathbf{z} \in \mathbb{R}^p$, *we have:*

$$\left\|\left(\frac{1}{n}\mathbf{X}^\mathsf{T}\mathbf{X} - \mathbf{I}\right)\mathbf{z}\right\|_\infty \leq k\mu\|\mathbf{z}\|_\infty.$$

**Lemma 3.** *Let* $\frac{1}{\sqrt{n}}\mathbf{X}$ *be a* $n \times p$ *matrix with* $\ell_2$-*normalized columns. Let* $\boldsymbol{\xi} \in \mathbb{R}^n$ *be a vector of independent* $\sigma^2$-*sub-Gaussian random variables. Then, with probability at least* $1 - \frac{1}{8p^3}$

$$\left\|\frac{1}{n}\mathbf{X}^\mathsf{T}\boldsymbol{\xi}\right\|_\infty \lesssim \sqrt{\frac{\sigma^2\log p}{n}}.$$

## A.4   Proof for Non-negative Signals

Recall the notation

$$\Phi(w_{\max}^\star, w_{\min}^\star, \epsilon, N) := \left(\frac{1}{8}\right)^{2/(N-2)} \wedge \left(\frac{(w_{\max}^\star)^{(N-2)/N}}{\log \frac{w_{\max}^\star}{\epsilon}}\right)^{2/(N-2)} \wedge \left(\frac{(w_{\min}^\star)^{(N-2)/N}}{\log \frac{w_{\min}^\star}{\epsilon}}\right)^{4/(N-2)},$$

and

$$\zeta := \frac{1}{5}w_{\min}^\star \vee \frac{200}{n}\left\|\mathbf{X}^\mathsf{T}\boldsymbol{\xi}\right\|_\infty \vee 200\epsilon.$$

**Theorem 3.** *Suppose that $\mathbf{w}^\star \succcurlyeq 0$ with $k \geq 1$ and $\mathbf{X}/\sqrt{n}$ satisfies $\mu$-incoherence with $\mu \leq C_\gamma/kr$, where $C_\gamma$ is some small enough constant. Take any precision $\epsilon > 0$, and let the initialization be such that*

$$0 < \alpha \leq \left(\frac{\epsilon}{p+1}\right)^{4/N} \wedge \Phi(w_{\max}^\star, w_{\min}^\star, \epsilon, N)$$

*For any iteration $t$ that satisfies*

$$\frac{1}{\eta N^2 \zeta^{(2N-2)/N}\alpha^{N-2}} \lesssim t \lesssim \frac{1}{\eta N^2 \tau}\left(\frac{1}{\alpha^{N-2}} - \frac{1}{\zeta^{(N-2)/2}}\right),$$

*the gradient descent algorithm (15) with step size $\eta \leq \frac{\alpha^N}{8N^2(w_{\max}^\star)^{(3N-2)/N}}$ yields the iterate $\mathbf{w}_t$ with the following property:*

$$|w_{t,i} - w_i^\star| \lesssim \begin{cases} \left\|\frac{1}{n}\mathbf{X}^\mathsf{T}\boldsymbol{\xi}\right\|_\infty \vee \epsilon & \text{if } i \in S \text{ and } w_{\min}^\star \lesssim \left\|\frac{1}{n}\mathbf{X}^\mathsf{T}\boldsymbol{\xi}\right\|_\infty \vee \epsilon, \\ \left|\frac{1}{n}(\mathbf{X}^\mathsf{T}\boldsymbol{\xi})_i\right| \vee k\mu\left\|\frac{1}{n}\mathbf{X}^\mathsf{T}\boldsymbol{\xi} \odot \mathbf{1}_S\right\|_\infty \vee \epsilon & \text{if } i \in S \text{ and } w_{\min}^\star \gtrsim \left\|\frac{1}{n}\mathbf{X}^\mathsf{T}\boldsymbol{\xi}\right\|_\infty \vee \epsilon, \\ \alpha^{N/4} & \text{if } i \notin S. \end{cases}$$

$$(16)$$

*Proof.* Let

$$\zeta := \frac{1}{5}w_{\min}^\star \vee \frac{2}{C_b}\left\|\frac{1}{n}\mathbf{X}^\mathsf{T}\boldsymbol{\xi}\right\|_\infty \vee \frac{2}{C_b}\epsilon,$$

where $C_b$ is some small enough positive constant that will be explicitly derived later. Also by the requirement of the coherence of the design matrix, we have

$$\|\mathbf{p}_t\|_\infty \leq \frac{C_\gamma}{w_{\max}^\star/w_{\min}^\star}\|\mathbf{s}_t - \mathbf{w}^\star\|_\infty.$$

Setting

$$\alpha \leq \left(\frac{\epsilon}{p+1}\right)^{4/N} \wedge \left(\frac{1}{8}\right)^{2/(N-2)} \wedge \left(\frac{(w_{\max}^\star)^{(N-2)/N}}{\log \frac{w_{\max}^\star}{\epsilon}}\right)^{2/(N-2)} \wedge \left(\frac{(w_{\min}^\star)^{(N-2)/N}}{\log \frac{w_{\min}^\star}{\epsilon}}\right)^{4/(N-2)}.$$

As long as $\|\mathbf{e}_t\|_\infty \leq \alpha^{N/4}$ we have

$$\begin{aligned}
\|\mathbf{b}_t\|_\infty + \alpha^{N/4} &\leq \left\|\frac{1}{n}\mathbf{X}^\mathsf{T}\boldsymbol{\epsilon}\right\|_\infty + \left\|\frac{1}{n}\mathbf{X}^\mathsf{T}\mathbf{X}\mathbf{e}_t\right\|_\infty + \alpha^{N/4} \\
&\leq 2\left(\left\|\frac{1}{n}\mathbf{X}^\mathsf{T}\boldsymbol{\epsilon}\right\|_\infty \vee (p\|\mathbf{e}_t\|_\infty) + \alpha^{N/4)}\right) \\
&\leq 2\left(\left\|\frac{1}{n}\mathbf{X}^\mathsf{T}\boldsymbol{\epsilon}\right\|_\infty \vee (p+1)\alpha^{N/4}\right) \\
&\leq C_b\frac{2}{C_b}\left(\left\|\frac{1}{n}\mathbf{X}^\mathsf{T}\boldsymbol{\epsilon}\right\|_\infty \vee \epsilon\right) \\
&\leq C_b\zeta.
\end{aligned}$$

where the second inequality is from Lemma 1. Further by Lemma 2, we also have

$$\|\mathbf{p}_t\|_\infty \le \frac{C_\gamma}{w_{\max}^\star / w_{\min}^\star} \|\mathbf{s}_t - \mathbf{w}^\star\|_\infty .$$

Therefore, both sequences $(\mathbf{b}_t)_{t \ge 0}$ and $(\mathbf{p}_t)_{t \ge 0}$ satisfy the assumptions of Proposition 1 conditionally on $\|\mathbf{e}_t\|_\infty$ staying below $\alpha^{N/4}$. If $\zeta \ge w_{\max}^\star$, at $t = 0$, we have already have

$$\|\mathbf{s}_0 - \mathbf{w}^\star\| \le \zeta.$$

Otherwise, applying Proposition 1, after

$$T_1 = \frac{75}{16\eta N^2 \zeta^{(2N-2)/N}} \log \frac{|w_{\max}^\star - \alpha^N|}{\epsilon} + \frac{15}{8N(N-2)\eta\zeta\alpha^{(N-2)}},$$

iterations and before

$$T_2 = \frac{5}{N(N-1)\eta\zeta} \left( \frac{1}{\alpha^{(N-2)}} - \frac{1}{\alpha^{(N-2)/2}} \right)$$

iterations, we have

$$\|\mathbf{s}_{T_1} - \mathbf{w}^\star\| \le \zeta,$$
$$\|\mathbf{e}_{T_1}\|_\infty \le \alpha^{N/2}.$$

If $\frac{1}{5} w_{\min}^\star \le \frac{2}{C_b} \left\| \frac{1}{n} \mathbf{X}^\mathsf{T} \boldsymbol{\xi} \right\|_\infty \vee \frac{2}{C_b}\epsilon$, then we are done.

If $\frac{1}{5} w_{\min}^\star > \frac{2}{C_b} \left\| \frac{1}{n} \mathbf{X}^\mathsf{T} \boldsymbol{\xi} \right\|_\infty \vee \frac{2}{C_b}\epsilon$, we have $\zeta = \frac{1}{5} w_{\min}^\star$. Choose $C_b + C_\gamma \le \frac{1}{40}$ as we have in Proposition 1. After $T_1$ iterations, we have

$$\|\mathbf{b}_t\|_\infty + \|\mathbf{p}_t\|_\infty \le C_b \frac{1}{5} w_{\min}^\star + \frac{C_\gamma}{w_{\max}^\star / w_{\min}^\star} \frac{1}{5} w_{\min}^\star \le (C_b + C_\gamma) \frac{1}{5} w_{\min}^\star \le \frac{1}{200} w_{\min}^\star.$$

Now all the assumptions of Proposition 2 are satisfied. To further reduce $\|\mathbf{s}_t - \mathbf{w}^\star\|_\infty$ from $\frac{1}{5} w_{\min}^\star$ to $\mathcal{O}(\left\| \frac{1}{n} \mathbf{X}^\mathsf{T} \boldsymbol{\xi} \right\|)$, we apply Proposition 2 and obtain that after

$$T_3 = \frac{6}{\eta N^2 (w_{\min}^\star)^{(2N-2)/N}} \log \frac{w_{\min}^\star}{\epsilon}$$

iterations and before

$$T_4 = \frac{25}{N(N-1)\eta w_{\min}^\star} \left( \frac{1}{\alpha^{(N-2)/2}} - \frac{1}{\alpha^{(N-2)/4}} \right)$$

iterations, we have for any $i \in S$,

$$|s_{t,i} - w_i^\star| \lesssim k\mu \max_{j \in S} B_j \vee B_i \vee \epsilon,$$
$$\|\mathbf{e}_t\|_\infty \le \alpha^{N/4}.$$

We use $\mathbb{1}\{\cdot\}$ to denote the indicator function. Therefore, the total number of iterations needed is

$$
\begin{aligned}
T_1 + T_3 = {} & \frac{75}{16\eta N^2 \zeta^{(2N-2)/N}} \log \frac{|w_{\max}^\star - \alpha^N|}{\epsilon} + \frac{15}{8N(N-2)\eta\zeta\alpha^{(N-2)}} \\
& + \frac{6}{\eta N^2 (w_{\min}^\star)^{(2N-2)/N}} \log \frac{w_{\min}^\star}{\epsilon} \mathbb{1}\left\{ \frac{1}{5} w_{\min}^\star > \frac{2}{C_b} \left\| \frac{1}{n} \mathbf{X}^\mathsf{T} \boldsymbol{\xi} \right\|_\infty \vee \frac{2}{C_b}\epsilon \right\}
\end{aligned}
\tag{17}
$$

and the upper bound for the total number of iterations would be

$$
\begin{aligned}
T_2 + T_4 = {} & \frac{5}{N(N-1)\eta\zeta} \left( \frac{1}{\alpha^{(N-2)}} - \frac{1}{\alpha^{(N-2)/2}} \right) \\
& + \frac{25}{N(N-1)\eta w_{\min}^\star} \left( \frac{1}{\alpha^{(N-2)/2}} - \frac{1}{\alpha^{(N-2)/4}} \right) \mathbb{1}\left\{ \frac{1}{5} w_{\min}^\star > \frac{2}{C_b} \left\| \frac{1}{n} \mathbf{X}^\mathsf{T} \boldsymbol{\xi} \right\|_\infty \vee \frac{2}{C_b}\epsilon \right\}
\end{aligned}
\tag{18}
$$

$\square$

# B  Multiplicative Update Sequences with General Order $N$

In this section, we analyze the one-dimensional updates that exhibits the similar dynamics to our gradient descent algorithm. The lemmas we derive will be assembled together to prove Proposition 1 and 2. The whole framework is similar to [11]. However, the continuous approximation plays an important role to deal with $N > 2$, and the detailed derivation differs from [11] a lot, especially for Lemma 5, 8 and 15.

## B.1  Error Growth

**Lemma 4.** *Consider the setting of updates given in equations* (14). *Suppose that* $\|\mathbf{e}_t\|_\infty \leq \frac{1}{8} w^\star_{min}$ *and there exists some* $B \in \mathbb{R}$ *such that for all $t$ we have* $\|\mathbf{b}_t\|_\infty + \|\mathbf{p}_t\|_\infty \leq B$. *Then, if* $\eta \leq \frac{1}{12(w^\star_{\max}+B)}$ *for any $t \geq 0$ we have*

$$\|\mathbf{e}_t\|_\infty \leq \|\mathbf{e}_0\|_\infty \prod_{i=1}^{t-1}(1 + 2N\eta(\|\mathbf{b}_i\|_\infty + \|\mathbf{p}_i\|_\infty)\|\mathbf{e}_i\|_\infty^{(N-2)/N})^N$$

*or in the other form,*

$$\|\mathbf{e}_{t+1}\|_\infty \leq \|\mathbf{e}_t\|_\infty (1 + 2N\eta(\|\mathbf{b}_t\|_\infty + \|\mathbf{p}_t\|_\infty)\|\mathbf{e}_t\|_\infty^{(N-2)/N})^N.$$

*Proof.* From the equations above, we get

$$\mathbf{1}_{S^c} \odot \mathbf{e}_{t+1} = \mathbf{1}_{S^c} \odot \mathbf{w}_t \odot (\mathbf{1} - 2N\eta(\mathbf{s}_t - \mathbf{w}^\star + \mathbf{p}_t + \mathbf{b}_t) \odot \mathbf{w}_t^{(N-2)/N})^N$$

$$= \mathbf{1}_{S^c} \odot \mathbf{e}_t \odot (\mathbf{1}_{S^c} - \mathbf{1}_{S^c} 2N\eta(\mathbf{s}_t - \mathbf{w}^\star + \mathbf{p}_t + \mathbf{b}_t) \odot \mathbf{e}_t^{(N-2)/N})^N$$

$$= \mathbf{1}_{S^c} \odot \mathbf{e}_t \odot (\mathbf{1} - 2N\eta(\mathbf{p}_t + \mathbf{b}_t) \odot \mathbf{e}_t^{(N-2)/N})^N$$

and hence

$$\|\mathbf{1}_{S^c} \odot \mathbf{e}_{t+1}\|_\infty \leq \|\mathbf{e}_t\|_\infty (1 + 2N\eta(\|\mathbf{b}_t\|_\infty + \|\mathbf{p}_t\|_\infty)\|\mathbf{e}_t\|_\infty^{(N-2)/N})^N.$$

$\square$

When we have the bound for $\|\mathbf{b}_t\|_\infty + \|\mathbf{p}_t\|_\infty$, we can control the size of $\|\mathbf{e}_t\|_\infty$ by the following lemma.

**Lemma 5.** *Let* $(b_t)_{t\geq 0}$ *be a sequence such that for $t \geq 0$ we have* $|b_t| \leq B$ *for some* $B > 0$. *Let the step size satisfy* $\eta \leq \frac{1}{4N(N-1)Bx_0^{(N-2)/(2N)}}$ *and consider a one-dimensional sequence* $(x_t)_{t\geq 0}$ *given by*

$$0 < x_0 < 1,$$

$$x_{t+1} = x_t(1 + 2N\eta b_t x_t^{(N-2)/N})^N.$$

*Then for any* $t < \frac{1}{8N(N-1)\eta B}\left(\frac{1}{x_0^{(N-2)/N}} - \frac{1}{x_0^{(N-2)/2N}}\right)$ *we have*

$$x_t \leq \sqrt{x_0}.$$

*Proof.* We start with studying the larger increasing rate of the updates,

$$x_{t+1} = x_t(1 + 2N\eta b_t x_t^{(N-2)/N})^N$$

$$\leq x_t(1 + 2N\eta B x_t^{(N-2)/N})^N$$

$$\leq x_t\left(1 + \frac{2N^2\eta B x_t^{(N-2)/N}}{1 - 2(N-1)N\eta x_t^{(N-2)/N}}\right)$$

$$\leq x_t(1 + 4N^2\eta B x_t^{(N-2)/N}),$$

where the second inequality is obtained by $(1+x)^r \leq 1 + \frac{rx}{1-(r-1)x}$ for $x \in (0, \frac{1}{r-1})$, and the last inequality is by the requirement of step size $\eta$. Therefore, to achieve to some value $x_T$, the number of iterations needed is lower bounded as

$$T \geq \sum_{t=0}^{T-1} \frac{x_{t+1} - x_t}{4N^2 \eta B x_t^{(2N-2)/N}}.$$

We aim at the number of iterations for $\sqrt{x_0}$, and we denote $T$ as the maximal number of iterations, i.e. $x_T < \sqrt{x_0}$ and $x_{T+1} \geq \sqrt{x_0}$. Therefore,

$$\frac{\sqrt{x_0} - x_T}{4N^2 \eta B x_T^{(2N-2)/N}} \leq \frac{x_{T+1} - x_T}{4N^2 \eta B x_T^{(2N-2)/N}} \leq 1.$$

And for $T$, we derive the lower bound as

$$\begin{aligned}
T \geq \sum_{t=0}^{T-1} \frac{x_{t+1} - x_t}{4N^2 \eta B x_t^{(2N-2)/N}} &\geq \frac{1}{4N^2 \eta B} \sum_{t=0}^{T-1} \int_{x_t}^{x_{t+1}} \frac{1}{x^{(2N-2)/N}} dx \\
&\geq \frac{1}{4N^2 \eta B} \int_{x_0}^{x_T} \frac{1}{x^{(2N-2)/N}} dx \\
&\geq \frac{1}{4N^2 \eta B} \int_{x_0}^{\sqrt{x_0}} \frac{1}{x^{(2N-2)/N}} dx - \frac{1}{4N^2 \eta B} \int_{x_T}^{\sqrt{x_0}} \frac{1}{x^{(2N-2)/N}} dx \\
&> \frac{1}{4N^2 \eta B} \left( -\frac{N}{2N-2} \frac{1}{x^{(N-2)/N}} \right) \Big|_{x_0}^{\sqrt{x_0}} - 1 \\
&= \frac{1}{8N(N-1)\eta B} \left( \frac{1}{x_0^{(N-2)/N}} - \frac{1}{x_0^{(N-2)/2N}} \right) - 1.
\end{aligned}$$

Therefore, we know that for any $t \leq \frac{1}{8N(N-1)\eta B} \left( \frac{1}{x_0^{(N-2)/N}} - \frac{1}{x_0^{(N-2)/2N}} \right) - 1$, we have $x_t \leq \sqrt{x_0}$. Since in practice $t$ is chosen as an integer, without loss of generality, we simply the requirement as

$$t < \frac{1}{8N(N-1)\eta B} \left( \frac{1}{x_0^{(N-2)/N}} - \frac{1}{x_0^{(N-2)/2N}} \right). \qquad \square$$

### B.2   Understanding 1-d Case

#### B.2.1   Basic Setting

In this subsection we analyze one-dimensional sequences with positive target corresponding to gradient descent updates without any perturbations. That is, $\mathbf{w}_t = \mathbf{u}_t^N$, $\frac{1}{n}\mathbf{X}^\top \mathbf{X} = \mathbf{I}$ and ignoring the error sequences $(\mathbf{b}_t)_{t \geq 0}$ and $(\mathbf{p}_t)_{t \geq 0}$. Hence, we will look at one-dimensional sequences of the form

$$\begin{aligned}
0 &< x_0 = \alpha^N < x^\star \\
x_{t+1} &= x_t (1 - 2N\eta(x_t - x^\star)x_t^{(N-2)/N})^N.
\end{aligned} \tag{19}$$

**Lemma 6** (Iterates behave monotonically). *Let $\eta > 0$ be the step size and suppose the updates are given by*

$$x_{t+1} = x_t (1 - 2N\eta(x_t - x^\star)x_t^{(N-2)/N})^N.$$

*Then the following holds*

1. *If $0 < x_0 \leq x^\star$ and $\eta \leq \frac{1}{2N(2N-2)(x^\star)^{(2N-2)/N}}$ then for any $t > 0$ we have $x_0 \leq x_{t-1} \leq x_t \leq x^\star$.*

2. *If $x^\star \leq x_0 \leq \frac{3}{2}x^\star$ and $\eta \leq \frac{1}{6N^2(x^\star)^{(2N-2)/N}}$ then for any $t \geq 0$ we have $x^\star \leq x_t \leq x_{t-1} \leq \frac{3}{2}x^\star$.*

*Proof.* Note that if $x_0 \leq x_t \leq x^\star$ then $x_t - x^\star \leq 0$ and hence $x_{t+1} \geq x_t$. Thus for the first part it is enough to show that for all $t \geq 0$ we have $x_t \leq x \leq x^\star$.

Assume for a contradiction that exists $t$ such that

$$x_0 \leq x_t \leq x^\star,$$
$$x_{t+1} > x^\star.$$

Plugging in the update rule for $x_{t+1}$ we can rewrite the above as

$$x_t \leq x^\star$$
$$< x_t(1 - 2N\eta(x_t - x^\star)x_t^{(N-2)/N})^N$$
$$\leq x_t \left( 1 + \frac{1}{2N-2} - \frac{x_t^{(2N-2)/N}}{(2N-2)(x^\star)^{(2N-2)/N}} \right)^N$$

Letting $\lambda = \left(\frac{x_t}{x^\star}\right)^{(2N-2)/N}$, by our assumption we have $0 < \lambda \leq 1$. The above inequality gives us

$$\left(\frac{1}{\lambda}\right)^{\frac{1}{2N-2}} < 1 + \frac{1}{2N-2} - \frac{1}{2N-2}\lambda.$$

And hence for $0 < \lambda \leq 1$ we have $f(\lambda) := \left(\frac{1}{\lambda}\right)^{\frac{1}{2N-2}} + \frac{1}{2N-2}\lambda < 1 + 1/(2N-2)$. Since for $0 < \lambda < 1$ we also have

$$f'(\lambda) = \frac{1}{2N-2} - \frac{1}{2N-2}\left(\frac{1}{\lambda}\right)^{\frac{1}{2N-2}+1} < 0,$$

so $f(\lambda) \geq f(1) = 1 + 1/(2N-2)$. This gives us the desired contradiction and concludes our proof for the first part.

We will now prove the second part. Similarly to the first part, we just need to show that for all $t \geq 0$ we have $x_t \geq x^\star$. Suppose that $x^\star \leq x_t \leq \frac{3}{2}x^\star$ and hence we can write $x_t = x^\star(1 + \gamma)$ for some $\gamma \in [0, \frac{1}{2}]$. Then we have

$$x_{t+1} = (1+\gamma)x^\star(1 - 2N\eta\gamma x^\star x_t^{(N-2)/N})^N$$
$$\geq (1+\gamma)x^\star(1 - 3N\eta\gamma(x^\star)^{(N-2)/N})^N$$
$$\geq x^\star(1+\gamma)\left(1 - \frac{1}{2N}\gamma\right)^N$$
$$\geq x^\star.$$

The last inequality is obtained by letting $f(\gamma) := (1 + \gamma)\left(1 - \frac{1}{2N}\gamma\right)^N$, we could get that

$$f'(\gamma) = \left(1 - \frac{1}{2N}\gamma\right)^N - \frac{1}{2}(1+\gamma)\left(1 - \frac{1}{2N}\gamma\right)^{N-1}$$
$$= \left(1 - \frac{1}{2N}\gamma\right)^{N-1}\left(\frac{1}{2} - \frac{1}{2}\gamma\right) > 0.$$

Hence, $f(\gamma) \geq f(0) = 1$ when $\gamma \in [0, \frac{1}{2}]$, which finishes the second part of our proof. $\square$

**Lemma 7** (Iterates behaviour near convergence). *Consider the same setting as before. Let $x^\star > 0$ and suppose that $|x_0 - x^\star| \leq \frac{1}{2}x^\star$. Then the following holds.*

1. *If $x_0 \leq x^\star$ and $\eta \leq \frac{1}{2N(2N-2)(x^\star)^{(2N-2)/N}}$, then for any $t \geq \frac{2}{\eta N^2 (x^\star)^{\frac{2N-2}{N}}}$ we have*

$$0 \leq x^\star - x_t \leq \frac{1}{2}|x_0 - x^\star|.$$

2. *If $x^\star \leq x_0 \leq \frac{3}{2}x^\star$ and $\eta \leq \frac{1}{6N^2(x^\star)^{(2N-2)/N}}$ then for any $t \geq \frac{1}{2N^2\eta(x^\star)^{(2N-2)/N}}$ we have*

$$0 \leq x_t - x^\star \leq \frac{1}{2}|x_0 - x^\star|.$$

*Proof.* Let us write $|x_0 - x^\star| = \gamma x^\star$ where $\gamma \in [0, \frac{1}{2}]$.

For the first part, we have $x_0 = (1 - \gamma)x^\star$, we want to know how many steps $t$ are needed to halve the error, i.e.,

$$x_t(1 - 2N\eta(x_t - x^\star)x_t^{\frac{N-2}{N}}))^N \geq (1 - \frac{\gamma}{2})x^\star.$$

We have that

$$x_t(1 - 2N\eta(x_t - x^\star)x_t^{\frac{N-2}{N}}))^N \geq x_t(1 + 2N\eta\frac{\gamma}{2}x^\star((1 - \gamma)x^\star)^{\frac{N-2}{N}}))^N$$

$$\geq x_0(1 + N\eta\gamma(1 - \gamma)^{\frac{N-2}{N}}(x^\star)^{\frac{2N-2}{N}}))^{Nt}$$

It is enough to have

$$x_0(1 + N\eta\gamma(1 - \gamma)^{\frac{N-2}{N}}(x^\star)^{\frac{2N-2}{N}}))^{Nt} \geq (1 - \frac{\gamma}{2})x^\star$$

$$\Rightarrow (1 - \gamma)(1 + tN^2\eta\gamma(1 - \gamma)^{\frac{N-2}{N}}(x^\star)^{\frac{2N-2}{N}})) \geq (1 - \frac{\gamma}{2})$$

$$\Rightarrow t \geq \left(\frac{1 - \frac{\gamma}{2}}{1 - \gamma} - 1\right)\frac{1}{N^2\eta\gamma(1 - \gamma)^{\frac{N-2}{N}}(x^\star)^{\frac{2N-2}{N}}}$$

$$\Rightarrow t \geq \frac{1}{2(1 - \gamma)^{\frac{2N-2}{N}}N^2\eta(x^\star)^{\frac{2N-2}{N}}}$$

$$\Rightarrow t \geq \frac{2}{\eta N^2(x^\star)^{\frac{2N-2}{N}}}$$

The last step is by $\gamma \in [0, \frac{1}{2}]$, we could obtain that $\frac{1}{2(1-\gamma)^{\frac{2N-2}{N}}} \leq \frac{1}{2(1/2)^{\frac{2N-2}{N}}} \leq \frac{1}{2(1/2)^2} \leq 2$.
Therefore after $t \geq \frac{2}{\eta N^2(x^\star)^{\frac{2N-2}{N}}}$, the error is halved.

To deal with the second part, we write $x_0 = x^\star(1 + \gamma)$. We will use a similar approach as the one in the first part. If for some $x_t$ we have $x_t \leq (1 + \gamma/2)x^\star$ we would be done. If $x_t > x^\star(1 + \gamma/2)$ we have $x_{t+1} \leq x_t(1 - 2N\eta\frac{\gamma}{2}x^\star(x^\star)^{(N-2)/N})^N$. Therefore,

$$x_0(1 - 2N\eta\frac{\gamma}{2}x^\star(x^\star)^{(N-2)/N})^{Nt} \leq x^\star(1 + \gamma/2)$$

$$\Longleftrightarrow Nt\log(1 - N\eta\gamma(x^\star)^{(2N-2)/N}) \leq \log\frac{x^\star(1 + \gamma/2)}{x_0}$$

$$\Longleftrightarrow t \geq \frac{1}{N}\frac{\log\frac{x^\star(1+\gamma/2)}{x_0}}{\log(1 - N\eta\gamma(x^\star)^{(2N-2)/N})}.$$

We can deal with the term on the right hand side by noting that

$$\frac{1}{N}\frac{\log\frac{x^\star(1+\gamma/2)}{x_0}}{\log(1 - N\eta\gamma(x^\star)^{(2N-2)/N})} = \frac{1}{N}\frac{\log\frac{1+\gamma/2}{1+\gamma}}{\log(1 - N\eta\gamma(x^\star)^{(2N-2)/N})}$$

$$\leq \frac{1}{N}\frac{\left(\frac{1+\gamma/2}{1+\gamma} - 1\right)/\left(\frac{1+\gamma/2}{1+\gamma}\right)}{-N\eta\gamma(x^\star)^{(2N-2)/N}}$$

$$= \frac{1}{N}\frac{-\frac{\gamma}{2}/(1 + \frac{\gamma}{2})}{-N\eta\gamma(x^\star)^{(2N-2)/N}}$$

$$\leq \frac{1}{2N^2\eta(x^\star)^{(2N-2)/N}}$$

where the second line used $\log x \leq x - 1$ and $\log x \geq \frac{x-1}{x}$. Note that both logarithms are negative. $\qquad\square$

**Lemma 8** (Iterates at the beginning). *Consider the same setting as before. If $0 < x_0 \leq \frac{1}{2}x^\star$ and $\eta \leq \frac{x_0}{2N(2N-4)(x^\star)^{(3N-2)/N}}$, for any $t \geq \frac{3}{2N(N-2)\eta x^\star x_0^{(N-2)/N}}$, we will have $\frac{1}{2}x^\star \leq x_t \leq x^\star$.*

*Proof.* We need to find a lower-bound on time $T$ which ensures that $x_T \geq \frac{x^\star}{2}$. At any time $t$, we have

$$x_{t+1} = x_t(1 - 2N\eta(x_t - x^\star)x_t^{(N-2)/N})^N \geq x_t(1 - 2N^2\eta(x_t - x^\star)x_t^{(N-2)/N}).$$

$$x_{t+1} - x_t \geq -2N^2\eta(x_t - x^\star)x_t^{(2N-2)/N}$$

$$\frac{x_{t+1} - x_t}{2N^2\eta(x^\star - x_t)x_t^{(2N-2)/N}} \geq 1$$

$$\sum_{t=0}^{T-1} \frac{x_{t+1} - x_t}{2N^2\eta(x^\star - x_t)x_t^{(2N-2)/N}} \geq \sum_{t=0}^{T-1} 1 = T.$$

Therefore, for $t$ that is larger than the left hand side, we have $x_t \geq \frac{1}{2}x^\star$.

$$\sum_{t=0}^{T-1} \frac{x_{t+1} - x_t}{2N^2\eta(x^\star - x_t)x_t^{(2N-2)/N}} \leq \frac{1}{N^2\eta x^\star} \sum_{t=0}^{T-1} \frac{x_{t+1} - x_t}{x_t^{(2N-2)/N}}$$

$$= \frac{1}{N^2\eta x^\star} \sum_{t=0}^{T-1} \int_{x_t}^{x_{t+1}} \frac{1}{x^{(2N-2)/N}} + \left( \frac{1}{x_t^{(2N-2)/N}} - \frac{1}{x^{(2N-2)/N}} \right) dx$$

$$\leq \frac{1}{N^2\eta x^\star} \sum_{t=0}^{T-1} \int_{x_t}^{x_{t+1}} \frac{1}{x^{(2N-2)/N}} dx$$

$$+ \frac{1}{N^2\eta x^\star} \max_{0 \leq t \leq T-1} \left( \frac{1}{x_t^{(2N-2)/N}} - \frac{1}{x_{t+1}^{(2N-2)/N}} \right) (x_T - x_0)$$

$$\leq \frac{1}{N^2\eta x^\star} \int_{x_0}^{\frac{1}{2}x^\star} \frac{1}{x^{(2N-2)/N}} dx \tag{20}$$

$$+ \frac{1}{N^2\eta x^\star} \max_{0 \leq t \leq T-1} \left( \frac{1}{x_t^{(2N-2)/N}} - \frac{1}{x_{t+1}^{(2N-2)/N}} \right) \left( \frac{1}{2}x^\star - x_0 \right) \tag{21}$$

$$+ \frac{1}{N^2\eta x^\star} \frac{1}{(\frac{1}{2}x^\star)^{(2N-2)/N}} \left( x_T - \frac{1}{2}x^\star \right) \tag{22}$$

For equation (20),

$$\frac{1}{N^2\eta x^\star} \int_{x_0}^{\frac{1}{2}x^\star} \frac{1}{x^{(2N-2)/N}} dx \leq \frac{1}{N^2\eta x^\star} \left( -\frac{N}{N-2} \frac{1}{x^{(N-2)/N}} \Big|_{x_0}^{\frac{1}{2}x^\star} \right)$$

$$= \frac{1}{N^2\eta x^\star} \left( -\frac{N}{N-2} \frac{1}{(\frac{1}{2}x^\star)^{(N-2)/N}} + -\frac{N}{N-2} \frac{1}{x_0^{(N-2)/N}} \right)$$

$$= \frac{1}{N(N-2)\eta x^\star} \left( \frac{1}{x_0^{(N-2)/N}} - \frac{2^{(N-2)/N}}{(x^\star)^{(N-2)/N}} \right).$$

For equation (21), we first focus on

$$\frac{1}{x_t^{(2N-2)/N}} - \frac{1}{x_{t+1}^{(2N-2)/N}}.$$

We have that

$$x_{t+1} = x_t(1 - 2N\eta(x_t - x^\star)x_t^{(N-2)/N})^N,$$

$$\Rightarrow x_{t+1}^{(2N-2)/N} = x_t^{(2N-2)/N}(1 - 2N\eta(x_t - x^\star)x_t^{(N-2)/N})^{2N-2}.$$

To deal with the multiplicative coefficient, with $\eta \leq \frac{1}{2N(2N-3)(x^\star)^{(2N-2)/N}}$ using the inequality $(1+x)^r \leq 1 + \frac{rx}{1-(r-1)x}$ where $x \in (0, \frac{1}{r-1})$, we obtain that

$$
\begin{aligned}
(1 - 2N\eta(x_t - x^\star)x_t^{(N-2)/N})^{2N-2} &\leq (1 + 2N\eta(x^\star)^{(2N-2)/N})^{(2N-2)} \\
&\leq 1 + \frac{2N(2N-2)\eta(x^\star)^{(2N-2)/N}}{1 - 2N(2N-3)\eta(x^\star)^{(2N-2)/N}} \\
&= \frac{1 - 2N\eta(x^\star)^{(2N-2)/N}}{1 - 2N(2N-3)\eta(x^\star)^{(2N-2)/N}}.
\end{aligned}
$$

Therefore,

$$
\begin{aligned}
\frac{1}{x_t^{(2N-2)/N}} - \frac{1}{x_{t+1}^{(2N-2)/N}} &= \frac{1}{x_t^{(2N-2)/N}} - \frac{1}{x_t^{(2N-2)/N}(1 - 2N\eta(x_t - x^\star)x_t^{(N-2)/N})^{2N-2}} \\
&= \frac{1}{x_t^{(2N-2)/N}}\left(1 - \frac{1}{(1 - 2N\eta(x_t - x^\star)x_t^{(N-2)/N})^{2N-2}}\right) \\
&\leq \frac{1}{x_t^{(2N-2)/N}}\left(1 - \frac{1 - 2N(2N-3)\eta(x^\star)^{(2N-2)/N}}{1 - 2N\eta(x^\star)^{(2N-2)/N}}\right) \\
&\leq \frac{1}{x_t^{(2N-2)/N}}\frac{2N(2N-4)\eta(x^\star)^{(2N-2)/N}}{1 - 2N\eta(x^\star)^{(2N-2)/N}} \\
&\leq \frac{1}{x_t^{(2N-2)/N}}2N(2N-4)\eta(x^\star)^{(2N-2)/N} \\
&\leq \frac{1}{x_0^{(2N-2)/N}}2N(2N-4)\eta(x^\star)^{(2N-2)/N}.
\end{aligned}
$$

If we further require the step size satisfies $\eta \leq \frac{x_0}{2N(2N-4)(x^\star)^{(3N-2)/N}}$, we have for equation (21),

$$
\begin{aligned}
\frac{1}{N^2\eta x^\star}\max_{0 \leq t \leq T-1}\left(\frac{1}{x_t^{(2N-2)/N}} - \frac{1}{x_{t+1}^{(2N-2)/N}}\right)\left(\frac{1}{2}x^\star - x_0\right) &\leq \frac{1}{N^2\eta x^\star}\frac{1}{x_0^{(N-2)/N}x^\star}\left(\frac{1}{2}x^\star - x_0\right) \\
&\leq \frac{1}{2N^2\eta x^\star}\frac{1}{x_0^{(N-2)/N}},
\end{aligned}
$$

which is with the same order with the result of equation (20).

Combining the results from equations (20), (21), (22), we obtain that

$$
\begin{aligned}
T &\leq \frac{1}{N(N-2)\eta x^\star}\left(\frac{1}{x_0^{(N-2)/N}} - \frac{2^{(N-2)/N}}{(x^\star)^{(N-2)/N}}\right) + \frac{1}{2N^2\eta x^\star}\frac{1}{x_0^{(N-2)/N}} \\
&\quad + \frac{1}{N^2\eta x^\star}\frac{1}{(\frac{1}{2}x^\star)^{(2N-2)/N}}\left(x_T - \frac{1}{2}x^\star\right) \\
&\leq \frac{1}{N(N-2)\eta x^\star}\left(\frac{1}{x_0^{(N-2)/N}} - \frac{2^{(N-2)/N}}{(x^\star)^{(N-2)/N}} + \frac{1}{2x_0^{(N-2)/N}} + \frac{1}{(\frac{1}{2}x^\star)^{(N-2)/N}}\right) \\
&\leq \frac{3}{2N(N-2)\eta x^\star x_0^{(N-2)/N}}.
\end{aligned}
$$

$\square$

**Lemma 9** (Overall iterates). *Consider the same setting as before. Fix any $\epsilon > 0$.*

*1. If $\epsilon < |x^\star - x_0| \leq \frac{1}{2}x^\star$ and $\eta \leq \frac{1}{6N^2(x^\star)^{(2N-2)/N}}$ then for any $t \geq \frac{3}{\eta N^2(x^\star)^{\frac{2N-2}{N}}}\log\frac{|x^\star - x_0|}{\epsilon}$ we have*

$$|x^\star - x_t| \leq \epsilon.$$

2. If $0 < x_0 \leq \frac{1}{2}x^\star$ and $\eta \leq \frac{x_0}{2N(2N-4)(x^\star)^{(3N-2)/N}}$ then for any

$$t \geq \frac{3}{\eta N^2 (x^\star)^{\frac{2N-2}{N}}} \log \frac{|x^\star - x_0|}{\epsilon} + \frac{3}{2N(N-2)\eta x^\star x_0^{(N-2)/N}}$$

we have

$$x^\star - \epsilon \leq x_t \leq x^\star.$$

*Proof.* 1. To prove the first part we simply need apply Lemma 7 $\lceil \log_2 \frac{|x^\star - x_0|}{\epsilon} \rceil$ times. Hence after

$$\frac{2 \log_2 e}{\eta N^2 (x^\star)^{\frac{2N-2}{N}}} \log \frac{|x^\star - x_0|}{\epsilon} \leq \frac{3}{\eta N^2 (x^\star)^{\frac{2N-2}{N}}} \log \frac{|x^\star - x_0|}{\epsilon}$$

iterations we are done.

2. For the second part, we simply combine the results from the first part and Lemma 8, it is enough to choose $t$ larger than or equal to

$$\frac{3}{\eta N^2 (x^\star)^{\frac{2N-2}{N}}} \log \frac{|x^\star - x_0|}{\epsilon} + \frac{3}{2N(N-2)\eta x^\star x_0^{(N-2)/N}}.$$

$\square$

### B.2.2 Dealing with Bounded Errors $\mathbf{b}_t$

In this subsection we extend the previous setting to handle bounded error sequences $(\mathbf{b}_t)_{t\geq 0}$ such that for any $t \geq 0$ we have $\|\mathbf{b}_t\|_\infty \leq B$ for some $B \in \mathbb{R}$. That is, we look at the following updates

$$x_{t+1} = x_t(1 - 2N\eta(x_t - x^\star + b_t)x_t^{(N-2)/N})^N.$$

Surely, if $B \geq x^\star$, the convergence to $x^\star$ is not possible. Hence, we will require $B$ to be small enough, with a particular choice $B \leq \frac{1}{5}x^\star$. For a given $B$, we can only expect the sequence $(x_t)_{t\geq 0}$ to converge to $x^\star$ up to precision $B$. We would consider two extreme scenarios,

$$x_{t+1}^+ = x_t^+ (1 - 2N\eta(x_t^+ - (x^\star - B))(x_t^+)^{(N-2)/N})^N,$$
$$x_{t+1}^- = x_t^- (1 - 2N\eta(x_t^- - (x^\star + B))(x_t^-)^{(N-2)/N})^N.$$

**Lemma 10** (Squeezing iterates with bounded errors). *Consider the sequences $(x_t^-)_{t\geq 0}, (x_t)_{t\geq 0}$ and $(x_t^+)_{t\geq 0}$ as defined above with*

$$0 < x_0^- = x_0^+ = x_0 \leq x^\star + B$$

*If $\eta \leq \frac{1}{8N^2(x^\star)^{(2N-2)/N}}$ then for all $t \geq 0$*

$$0 \leq x_t^- \leq x_t \leq x_t^+ \leq x^\star + B.$$

*Proof.* We will prove the claim by induction. The claim holds trivially for $t = 0$. If $x_t^+ \geq x_t$, we have

$$x_{t+1}^+ = x_t^+ (1 - 2N\eta(x_t^+ - (x^\star + B))(x_t^+)^{\frac{N-2}{N}})^N$$

$$\geq x_t^+ (1 - 2N\eta(x_t^+ - (x^\star + B))x_t^{\frac{N-2}{N}})^N$$

$$(\triangle = x_t^+ - x_t) \quad = (x_t + \triangle)(1 - 2N\eta(x_t - x^\star + b_t)x_t^{\frac{N-2}{N}}$$

$$+ 2N\eta(x_t^+ - x_t - B - b_t)x_t^{\frac{N-2}{N}})^N$$

$$(m_t = 1 - 2N\eta(x_t - x^\star + b_t)x_t^{\frac{N-2}{N}}) \quad \geq (x_t + \triangle)(m_t - 2N\eta\triangle x_t^{\frac{N-2}{N}})^N$$

$$\geq (x_t + \triangle)(m_t - 2N\eta\triangle x_t^{\frac{N-2}{N}})^N$$

$$= x_t m_t^N + (x_t + \triangle)(m_t - 2N\eta\triangle x_t^{\frac{N-2}{N}})^N - x_t m_t^N$$

$$= x_t m_t^N + (x_t + \triangle)m_t^N \left(1 - \frac{2N\eta\triangle x_t^{\frac{N-2}{N}}}{m_t}\right)^N - x_t m_t^N.$$

We aimed to show that $(x_t + \triangle)m_t^N(1 - 2N\eta\triangle x_t^{\frac{N-2}{N}}/m_t)^N - x_t m_t^N$ is positive. With $\eta \leq \frac{1}{4N(x^\star+B)(x^\star)^{(N-2)/N}}$, we can see $m_t \geq 1/2$ for all $t$ and

$$(x_t + \triangle)m_t^N(1 - 2N\eta\triangle x_t^{\frac{N-2}{N}}/m_t)^N - x_t m_t^N \geq (x_t + \triangle)m_t^N(1 - 4N\eta\triangle x_t^{\frac{N-2}{N}})^N - x_t m_t^N$$
$$\geq (x_t + \triangle)m_t^N(1 - 4N^2\eta\triangle x_t^{\frac{N-2}{N}}) - x_t m_t^N.$$

The last inequality is obtained via $(1 - x)^n \geq 1 - nx$. If we further require $\eta \leq \frac{1}{8N^2(x^\star)^{(2N-2)/N}}$, we obtain that

$$(x_t + \triangle)m_t^N(1 - 4N^2\eta\triangle x_t^{\frac{N-2}{N}}) - x_t m_t^N \geq (x_t + \triangle)m_t^N\left(1 - \frac{1}{2x^\star}\triangle\right) - x_t m_t^N$$
$$\geq m_t^N\left(x_t + \triangle - \frac{x_t}{2x^\star}\triangle - \frac{1}{2x^\star}\triangle^2 - x_t\right)$$
$$\geq m_t^N\triangle\left(1 - \frac{x_t}{2x^\star} - \frac{\triangle}{2x^\star}\right)$$
$$\geq m_t^N\triangle\left(1 - \frac{1}{2} - \frac{1}{2}\right) \geq 0.$$

Therefore, we obtain that

$$x_{t+1}^+ \geq x_t m_t^N = x_{t+1}.$$

For $x_t^-$, it follows a similar proof. $\qquad\square$

**Lemma 11** (Iterates with bounded errors monotonic behaviour). *Consider the previous setting with* $B \leq \frac{1}{5}x^\star$, $\eta \leq \frac{1}{6N^2(x^\star)^{(2N-2)/N}}$. *Then the following holds*

1. *If* $|x_t - x^\star| > B$ *then* $|x_{t+1} - x^\star| < |x_t - x^\star|$.
2. *If* $|x_t - x^\star| \leq B$ *then* $|x_{t+1} - x^\star| \leq B$.

*Proof.* The choice of $B$ and step size $\eta$ ensures us to apply Lemma 6 and Lemma 10 to the sequences $(x_t^-)_{t\geq0}$ and $(x_t^+)_{t\geq0}$. $\qquad\square$

**Lemma 12** (Iterates with $B$ near convergence). *Consider the setting as before. Then the following holds:*

1. *If* $\frac{1}{2}(x^\star - B) \leq x_0 \leq x^\star - 5B$ *then for any* $t \geq \frac{2}{\eta N^2(x^\star)^{\frac{2N-2}{N}}}$ *we have*

$$|x^\star - x_t| \leq \frac{1}{2}|x_0 - x^\star|.$$

2. *If* $x^\star + 4B < x_0 < \frac{6}{5}x^\star$ *then for any* $t \geq \frac{4}{\eta N^2(x^\star)^{\frac{2N-2}{N}}}$ *we have*

$$|x^\star - x_t| \leq \frac{1}{2}|x_0 - x^\star|.$$

*Proof.* 1. To prove the first part, let us first apply Lemma 7 on $x_t^-$ twice, therefore for all

$$t \geq \frac{25}{4\eta N^2(x^\star)^{\frac{2N-2}{N}}} \geq 2\frac{2}{\eta N^2(x^\star - B)^{\frac{2N-2}{N}}}$$

we have

$$0 \leq (x^\star - B) - x_t^-$$
$$\leq \frac{1}{4}|x_0 - (x^\star - B)|$$
$$\leq \frac{1}{4}|x_0 - x^\star| + \frac{1}{4}B.$$

When $x_t \leq x^\star$, from Lemma 10 we have

$$
\begin{aligned}
0 &\leq x^\star - x_t \\
&\leq x^\star - x_t^- \\
&\frac{1}{4}|x_0 - x^\star| + \frac{5}{4}B \\
&\leq \frac{1}{2}|x_0 - x^\star|.
\end{aligned}
$$

When $x_t \geq x^\star$ then by Lemma 10 we have

$$
0 \leq x_t - x^\star \leq B \leq \frac{1}{5}|x_0 - x^\star|,
$$

where both last inequalities are from $x_0 \leq x^\star - 5B$.

2. The second part follows a very similar proof for $x_t^+$, the number of iterations would be

$$
t \geq \frac{4}{\eta N^2 (x^\star)^{\frac{2N-2}{N}}} \geq 2 \frac{2}{\eta N^2 (x^\star + B)^{\frac{2N-2}{N}}}.
$$

$\square$

**Lemma 13** (Overall iterates with $B$). *Consider the same setting as before. Fix any $\epsilon > 0$, then the following holds*

1. *If $B + \epsilon < |x^\star - x_0| \leq \frac{1}{5}x^\star$ then for any $t \geq \frac{15}{4\eta N^2 (x^\star)^{\frac{2N-2}{N}}} \log \frac{|x^\star - x_0|}{\epsilon}$ iterations we have $|x^\star - x_t| \leq B + \epsilon$.*

2. *If $0 < x_0 \leq x^\star - B - \epsilon$ then for any*

$$
t \geq \frac{75}{16\eta N^2 (x^\star)^{\frac{2N-2}{N}}} \log \frac{|x^\star - x_0|}{\epsilon} + \frac{15}{8N(N-2)\eta x^\star x_0^{(N-2)/N}}
$$

*we have $x^\star - B - \epsilon \leq x_t \leq x^\star + B$.*

*Proof.*     1. If $x_0 > x^\star + B$ then by Lemma 10 and Lemma 11 we only need to show that $(x_t^+)_{t \geq 0}$ hits $x^\star + B + \epsilon$ within the desired number of iterations. From the first part of Lemma 9, we see that

$$
\frac{3}{\eta N^2 (x^\star + B)^{\frac{2N-2}{N}}} \log \frac{|x^\star + B - x_0|}{\epsilon} \leq \frac{15}{4\eta N^2 (x^\star)^{\frac{2N-2}{N}}} \log \frac{|x^\star - x_0|}{\epsilon}
$$

iterations are enough, where we require $\frac{|x^\star - x_0|}{\epsilon} \geq \frac{5}{2}$.

2. The upper bound is obtained immediately from Lemma 10. For lower bound, we simply apply the second part of Lemma 9 to the sequence $(x_t^-)_{t \geq 0}$ to get

$$
\begin{aligned}
t &\geq \frac{75}{16\eta N^2 (x^\star)^{\frac{2N-2}{N}}} \log \frac{|x^\star - x_0|}{\epsilon} + \frac{15}{8N(N-2)\eta x^\star x_0^{(N-2)/N}} \\
&\geq \frac{3}{\eta N^2 (x^\star - B)^{\frac{2N-2}{N}}} \log \frac{|x^\star - B - x_0|}{\epsilon} + \frac{3}{2N(N-2)\eta (x^\star - B)x_0^{(N-2)/N}}
\end{aligned}
$$

to ensure the results we wanted.

$\square$

**Lemma 14.** *Suppose the error sequences* $(\mathbf{b}_t)_{t\geq 0}$ *and* $(\mathbf{p}_t)_{t\geq 0}$ *satisfy the following for any* $t \geq 0$:

$$\|\mathbf{b}_t \odot \mathbf{1}_S\| \leq B,$$

$$\|\mathbf{p}_t\|_\infty \leq \frac{1}{20} \|\mathbf{s}_t - \mathbf{w}^\star\|_\infty.$$

*Suppose that*

$$20B < \|\mathbf{s}_0 - \mathbf{w}^\star\|_\infty \leq \frac{1}{5}w^\star_{\min}.$$

*Then for* $\eta \leq \frac{1}{6N^2(w^\star_{\max})^{(2N-2)/N}}$ *and any* $t \geq \frac{2}{\eta N^2(w^\star_{\max})^{(2N-2)/N}}$ *we have*

$$\|\mathbf{s}_t - \mathbf{w}^\star\|_\infty \leq \frac{1}{2} \|\mathbf{s}_0 - \mathbf{w}^\star\|_\infty.$$

*Proof.* Note that $\|\mathbf{b}_0\|_\infty + \|\mathbf{p}_t\|_\infty \leq \frac{1}{10} \|\mathbf{s}_0 - \mathbf{w}^\star\|_\infty$. For any $i$ such that $|s_{0,i} - w^\star_i| \leq \frac{1}{2} \|\mathbf{s}_0 - \mathbf{w}^\star\|_\infty$, Lemma 11 guarantees that for any $t \geq 0$ we have $|s_{t,i} - w^\star_i| \leq \frac{1}{2} \|\mathbf{s}_0 - \mathbf{w}^\star\|_\infty$. On the other hand, for any $i$ such that $|s_{0,i} - w^\star_i| > \frac{1}{2} \|\mathbf{s}_0 - \mathbf{w}^\star\|_\infty$ by Lemma 12 we have $|s_{0,i} - w^\star_i| \leq \frac{1}{2} \|\mathbf{s}_0 - \mathbf{w}^\star\|_\infty$ for any $t \geq \frac{2}{\eta N^2(w^\star_{\max})^{(2N-2)/N}}$ which concludes the proof. $\qquad\square$

### B.3  Dealing with Negative Targets

**Lemma 15.** *Let* $x_t = u^N - v^N$ *and* $x^\star \in \mathbb{R}$ *be the target such that* $|x^\star| > 0$. *Suppose the sequences* $(u_t)_{t\geq 0}$ *and* $(v_t)_{t\geq 0}$ *evolve as follows*

$$0 < u_0 = \alpha, \quad u_{t+1} = u_t(1 - 2N\eta(x_t - x^\star + b_t)u_t^{N-2}),$$
$$0 < v_0 = \alpha, \quad v_{t+1} = v_t(1 + 2N\eta(x_t - x^\star + b_t)v_t^{N-2}),$$

*where* $\alpha \leq (2 - 2^{\frac{N-2}{N}})^{\frac{1}{N-2}}|x^\star|^{1/N}$ *and there exists* $B > 0$ *such that* $|b_t| \leq B$ *and* $\eta \leq \frac{\alpha}{4N(N-2)(x^\star+B)x^\star}$. *Then the following holds: For any* $t \geq 0$ *we have*

- *If* $x^\star > 0$ *and* $u_t^N \geq x^\star$, *then* $v_t^N \leq \frac{1}{2}\alpha^N$.

- *If* $x^\star < 0$ *and* $v_t^N \geq |x^\star|$, *then* $u_t^N \leq \frac{1}{2}\alpha^N$.

*Proof.* Let us assume $x^\star > 0$ first and prove the first statement. From the updating equation, we obtain that

$$\frac{u_{t+1} - u_t}{u_t^{N-1}} = -2N\eta(x_t - x^\star + b_t).$$

Therefore,

$$\sum_{i=0}^{t} -2N\eta(x_i - x^\star + b_i) = \sum_{i=0}^{t} \frac{u_{i+1} - u_i}{u_i^{N-1}}$$

$$\geq \sum_{i=0}^{t} \int_{u_i}^{u_{i+1}} \frac{1}{u^{N-1}} du$$

$$= \int_{u_0}^{u_t} \frac{1}{u^{N-1}} du$$

$$= (2 - N)(u_t^{2-N} - u_0^{2-N}).$$

When $u_t^N \geq x^\star$, we have that $u_t^{2-N} \leq (x^\star)^{(2-N)/N}$. Therefore,

$$\sum_{i=1}^{t} -2N\eta(x_i - x^\star + b_i) \geq (2 - N)(u_t^{2-N} - u_0^{2-N}).$$

Similarly for $v_t$, we have

$$\sum_{i=1}^{t} 2N\eta(x_i - x^\star + b_i) = \sum_{i=0}^{t} \frac{v_{i+1} - v_i}{v_i^{N-1}}$$
$$\geq (2-N)(v_t^{2-N} - v_0^{2-N}).$$

Therefore, we have that

$$(N-2)((x^\star)^{\frac{2-N}{N}} - \alpha^{2-N}) \geq (2-N)(v_t^{2-N} - \alpha^{2-N}).$$
$$\implies (\alpha^{2-N} - (x^\star)^{\frac{2-N}{N}}) + \alpha^{2-N} \leq v_t^{2-N}.$$
$$\implies v_t \leq \left(\frac{1}{2\alpha^{2-N} - (x^\star)^{\frac{2-N}{N}}}\right)^{\frac{1}{N-2}}$$
$$\implies v_t \leq \left(\frac{1}{2 - \alpha^{N-2}/(x^\star)^{\frac{N-2}{N}}}\right)^{\frac{1}{N-2}} \alpha$$
$$\implies v_t \leq \left(\frac{1}{2 - (2 - 2^{\frac{N-2}{N}})}\right)^{\frac{1}{N-2}} \alpha$$
$$\implies v_t \leq 2^{\frac{1}{N}} \alpha.$$

For $x^\star < 0$, we obtain a similar result by symmetry. $\qquad\square$

**Lemma 16.** *Let $x_t = x_t^+ - x_t^-$ and $x^\star \in \mathbb{R}$ be the target such that $|x^\star| > 0$. Suppose the sequences $(x_t^+)_{t\geq 0}$ and $(x_t^-)_{t\geq 0}$ evolve as follows*

$$0 < x_0^+ = \alpha^N \leq 2^3(2^{\frac{1}{N}} - 1)^{\frac{N}{N-2}}|x^\star|, \quad x_{t+1}^+ = x_t^+(1 - 2N\eta(x_t - x^\star + b_t)(x_t^+)^{(N-2)/N})^N,$$
$$0 < x_0^- = \alpha^N \leq 2^3(2^{\frac{1}{N}} - 1)^{\frac{N}{N-2}}|x^\star|, \quad x_{t+1}^- = x_t^-(1 + 2N\eta(x_t - x^\star + b_t)(x_t^-)^{(N-2)/N})^N,$$

*and that there exists $B > 0$ such that $|b_t| \leq B$ and $\eta \leq \frac{1}{8N(x^\star + B)(x^\star)^{(N-2)/N}}$. Then the following holds: For any $t \geq 0$ we have*

- *If $x^\star > 0$ then $x_t^- \leq \alpha^N \Pi_{i=0}^{t-1}(1 + 2N\eta|b_t|(x_i^-)^{(N-2)/N})^N$.*
- *If $x^\star < 0$ then $x_t^+ \leq \alpha^N \Pi_{i=0}^{t-1}(1 + 2N\eta|b_t|(x_i^+)^{(N-2)/N})^N$.*

*Proof.* Assume $x^\star > 0$ and fix any $t \geq 0$. Let $0 \leq s \leq t$ be the largest $s$ such that $x_s^+ > x^\star$. If no such $s$ exists we are done immediately. If $s = t$ then by the first part we have $x_t^- \leq \alpha^N$ and we are done.

If $s < t$, by Lemma 15, we have $x_s^- \leq \frac{1}{2}\alpha^N$. From the requirement of initialization, we have

$$(1 + 2N\eta(x_s^+ - x_s^- - x^\star + b_s)(x_s^-)^{(N-2)/N})^N \leq \left(1 + \frac{(\frac{1}{2}\alpha^N)^{\frac{N-2}{N}}}{4(x^\star)^{\frac{N-2}{N}}}\right)^N \leq (1 + 2^{\frac{1}{N}} - 1)^N = 2.$$

Therefore

$$x_t^- = x_s^- \prod_{i=s}^{t-1}(1 + 2N\eta(x_i^+ - x_i^- - x^\star + b_i)(x_i^-)^{(N-2)/N})^N$$
$$= \frac{1}{2}\alpha^N \cdot 2 \prod_{i=s+1}^{t-1}(1 + 2N\eta(x_i^+ - x_i^- - x^\star + b_i)(x_i^-)^{(N-2)/N})^N$$
$$\leq \alpha^N \prod_{i=s+1}^{t-1}(1 + 2N\eta|b_i|(x_i^-)^{(N-2)/N})^N.$$

This completes the proof for $x^\star > 0$. It follows a similar proof for the case $x^\star < 0$. □

# C   Proof of Propositions and Technical Lemmas

In this section, we provide the proof for the propositions and technical lemmas mentioned in Appendix A.

## C.1   Proof of Proposition 1

By the assumptions on $(\mathbf{b}_t)_{t\geq 0}$ and $(\mathbf{p}_t)_{t\geq 0}$, we obtain that

$$\|\mathbf{b}_t\|_\infty \leq C_b\zeta - \alpha^{N/4},$$

$$\|\mathbf{p}_t\|_\infty \leq \frac{C_\gamma}{w^\star_{\max}/\zeta}\|\mathbf{s}_t - \mathbf{w}^\star\|_\infty \leq \frac{C_\gamma}{w^\star_{\max}/\zeta}w^\star_{\max} \leq C_\gamma\zeta.$$

Choose $C_b$ and $C_\gamma$ such that $C_b + C_\gamma \leq 1/40$. Therefore, we have

$$B \leq \|\mathbf{b}_t\|_\infty + \|\mathbf{p}_t\|_\infty + \alpha^{N/4} \leq (C_b + C_\gamma)\zeta \leq \frac{1}{40}\zeta.$$

For any $j$ such that $w^\star_j \geq \frac{1}{2}\zeta$, we have that $B \leq \frac{1}{20}w^\star_j$. Therefore, by applying Lemma 13, we know when

$$t \geq \frac{75}{16\eta N^2\zeta^{(2N-2)/N}}\log\frac{|w^\star_{\max} - \alpha^N|}{\epsilon} + \frac{15}{8N(N-2)\eta\zeta\alpha^{(N-2)}} = T_1,$$

we have $|w_{j,t} - w^\star_j| \leq \zeta$.

On the other hand, for any $j$ such that $w^\star_j \leq \frac{1}{2}\zeta$, $w_{j,t}$ will stay in $(0, w^\star_j + \frac{1}{40}\zeta]$ maintaining $|w_{j,t} - w^\star_j| \leq \zeta$ as required.

By Lemma 5, we have that $\|\mathbf{e}_t\|_\infty \leq \alpha^{N/2}$ up to

$$T_2 = \frac{5}{N(N-1)\eta\zeta}\left(\frac{1}{\alpha^{N-2}} - \frac{1}{\alpha^{\frac{N-2}{2}}}\right).$$

From our choice of initialization $\alpha$, we can see that $T_1 \leq T_2$ is ensured. To see this,

$$\alpha \leq \left(\frac{1}{8}\right)^{2/(N-2)} \wedge \left(\frac{\zeta^{(N-2)/N}}{\log\frac{w^\star_{\max}}{\epsilon}}\right)^{2/(N-2)}$$

$$\Longrightarrow \alpha^{(N-2)/2} \leq \frac{1}{8} \wedge \frac{16\zeta^{(N-2)/N}}{15\log\frac{w^\star_{\max}}{\epsilon}}$$

$$\Longrightarrow 4\alpha^{(N-2)/2}\left(\frac{15}{16}\alpha^{(N-2)/2}\log\frac{w^\star_{\max}}{\epsilon} + \zeta^{(N-2)/N}\right) \leq \zeta^{(N-2)/N}$$

$$\Longrightarrow \frac{15}{2\zeta^{(N-2)/N}}\log\frac{w^\star_{\max}}{\epsilon} + \frac{6}{\alpha^{(N-2)}} \leq 8\left(\frac{1}{\alpha^{(N-2)}} - \frac{1}{\alpha^{(N-2)/2}}\right)$$

$$\Longrightarrow \frac{75}{16\eta N^2\zeta^{(2N-2)/N}}\log\frac{|w^\star_{\max} - \alpha^N|}{\epsilon} + \frac{15}{8N(N-2)\eta\zeta\alpha^{(N-2)}} \leq \frac{5}{N(N-1)\eta\zeta}\left(\frac{1}{\alpha^{(N-2)}} - \frac{1}{\alpha^{(N-2)/2}}\right)$$

$$\Longrightarrow T_1 \leq T_2.$$

$$\tag{23}$$

□

## C.2   Proof of Proposition 2

By Lemma 5, with the choice of $B = \frac{1}{200}w^\star_{\min}$, we can maintain $\|\mathbf{e}_t\|_\infty \leq \alpha^{N/4}$ for at least another

$$t \leq \frac{25}{N(N-1)\eta w^\star_{\min}}\left(\frac{1}{\alpha^{(N-2)/2}} - \frac{1}{\alpha^{(N-2)/4}}\right) = T_4.$$

Now we consider to further reduce $\|\mathbf{s}_t - \mathbf{w}^\star\|_\infty$ from $\frac{1}{5}w^\star_{\min}$ to $\left\|\frac{1}{n}\mathbf{X}^\top\boldsymbol{\xi}\right\|_\infty \vee \epsilon$. Let $B_i := (\mathbf{b}_t)_i$ and $B := \max_{j\in S} B_j$.

We first apply Lemma 14 for $\log_2 \frac{w^\star_{\min}}{100(B\vee\epsilon)}$ times, the total number of iterations for this step would be

$$\frac{2}{\eta N^2 (w^\star_{\min})^{(2N-2)/N}} \log_2 \frac{w^\star_{\min}}{100(B\vee\epsilon)}.$$

After that we have $\|\mathbf{s}_t - \mathbf{w}^\star\|_\infty < 20(B\vee\epsilon)$ and so $\|\mathbf{p}_t\|_\infty < k\mu \cdot 20(B\vee\epsilon)$. Hence, for any $i\in S$ we have

$$\|\mathbf{b}_t \odot \mathbf{1}_i\|_\infty + \|\mathbf{p}_t\|_\infty \le B_i + k\mu 20(B\vee\epsilon).$$

Then we further apply Lemma 13 for each coordinate $i\in S$ to obtain that

$$|w_{i,t} - w^\star_i| \lesssim \left|\frac{1}{n}(\mathbf{X}^\top\boldsymbol{\xi})_i\right| \vee k\mu \left\|\frac{1}{n}\mathbf{X}^\top\boldsymbol{\xi}\odot\mathbf{1}_S\right\|_\infty \vee \epsilon.$$

the number of iterations needed for this step is $\frac{15}{4\eta N^2(w^\star_{\min})^{(2N-2)/N}}\log\frac{w^\star_{\min}}{\epsilon}$.

Therefore the total number of iterations needed to further reduce $\|\mathbf{s}_t - \mathbf{w}^\star\|_\infty$ is

$$T_3 = \frac{6}{\eta N^2(w^\star_{\min})^{(2N-2)/N}}\log\frac{w^\star_{\min}}{\epsilon}$$

$$\ge \frac{2}{\eta N^2(w^\star_{\min})^{(2N-2)/N}}\log_2\frac{w^\star_{\min}}{100(B\vee\epsilon)} + \frac{15}{4\eta N^2(w^\star_{\min})^{(2N-2)/N}}\log\frac{w^\star_{\min}}{\epsilon}.$$

Since $T_3$ is no longer related to $\alpha$, we can easily ensure $T_3 \le T_4$ with some mild upper bound on $\alpha^{(N-2)/4} \le \frac{(w^\star_{\min})^{(N-2)/N}}{\log\frac{w^\star_{\min}}{\epsilon}} \wedge 1/2$.

$$\alpha^{(N-2)/4} \le \frac{(w^\star_{\min})^{(N-2)/N}}{\log\frac{w^\star_{\min}}{\epsilon}} \wedge 1/2$$

$$\Longrightarrow \alpha^{(N-2)/4}\left(\alpha^{(N-2)/4}\log\frac{w^\star_{\min}}{\epsilon} + (w^\star_{\min})^{(N-2)/N}\right) \le (w^\star_{\min})^{(N-2)/N}$$

$$\Longrightarrow \alpha^{(N-2)/2}\log\frac{w^\star_{\min}}{\epsilon} \le (w^\star_{\min})^{(N-2)/N} - \alpha^{(N-2)/4}(w^\star_{\min})^{(N-2)/N}$$

$$\Longrightarrow \frac{1}{(w^\star_{\min})^{(N-2)/N}}\log\frac{w^\star_{\min}}{\epsilon} \le \frac{1}{\alpha^{(N-2)/2}} - \frac{1}{\alpha^{(N-2)/4}} \tag{24}$$

$$\Longrightarrow \frac{6}{\eta N^2(w^\star_{\min})^{(2N-2)/N}}\log\frac{w^\star_{\min}}{\epsilon} \le \frac{25}{\eta N(N-1)w^\star_{\min}}\left(\frac{1}{\alpha^{(N-2)/2}} - \frac{1}{\alpha^{(N-2)/4}}\right)$$

$$\Longrightarrow T_3 \le T_4.$$

$\square$

## C.3 Proof of Technical Lemmas

*Proof of Lemma 1.* Since $\frac{1}{\sqrt{n}}\mathbf{X}$ is with $\ell_2$-normalized columns and satisfies $\mu$-coherence, where $0 \le \mu \le 1$,

$$\left|\left(\frac{1}{n}\mathbf{X}^\top\mathbf{X}\right)_{i,j}\right| = \left|\left(\frac{1}{\sqrt{n}}\mathbf{X}_i\right)^\top\left(\frac{1}{\sqrt{n}}\mathbf{X}_j\right)\right| \le \max\{1,\mu\} \le 1.$$

Therefore, for any $\mathbf{z} \in \mathbb{R}^p$,

$$\left\|\frac{1}{n}\mathbf{X}^\top\mathbf{X}\mathbf{z}\right\|_\infty \le p\|\mathbf{z}\|_\infty.$$

$\square$

*Proof of Lemma 2.* It is straightforward to verify that for any $i\in\{1,\ldots,p\}$,

$$\left|\left(\frac{1}{n}\mathbf{X}^\top\mathbf{X}\mathbf{z}\right)_i - \mathbf{z}_i\right| \le k\mu\|\mathbf{z}\|_\infty.$$

Therefore,

$$\left\|\left(\frac{1}{n}\mathbf{X}^\mathsf{T}\mathbf{X}-\mathbf{I}\right)\mathbf{z}\right\|_\infty \leq k\mu\left\|\mathbf{z}\right\|_\infty.$$

$\square$

*Proof of Lemma 3.* Since the vector $\boldsymbol{\xi}$ are made of independent $\sigma^2$-subGaussian random variables and any column $\mathbf{X}_i$ of $\mathbf{X}$ is $\ell_2$-normalized, i.e. $\left\|\frac{1}{\sqrt{n}}\mathbf{X}_i\right\|=1$, the random variable $\frac{1}{\sqrt{n}}(\mathbf{X}^\mathsf{T}\boldsymbol{\xi})_i$ is still $\sigma^2$-subGaussian.

It is a standard result that for any $\epsilon > 0$,

$$\mathbb{P}\left(\left\|\frac{1}{\sqrt{n}}\mathbf{X}^\mathsf{T}\boldsymbol{\xi}\right\|_\infty > \epsilon\right) \leq 2p\exp\left(-\frac{\epsilon^2}{2\sigma^2}\right).$$

Setting $\epsilon = 2\sqrt{2\sigma^2\log(2p)}$, with probability at least $1 - \frac{1}{8p^3}$ we have

$$\left\|\frac{1}{n}\mathbf{X}^\mathsf{T}\boldsymbol{\xi}\right\|_\infty \leq \frac{1}{\sqrt{n}}2\sqrt{\sigma^2\log(2p)} \lesssim \sqrt{\frac{\sigma^2\log p}{n}}.$$

$\square$

# D   Proof of Theorems in Section 3

In this section, we provide the proof for all results we mentioned in Section 3.

## D.1   Proof of Theorem 1

*Proof.* Now let us consider the updates in equation (14). The major idea is to show that the results in Theorem 3 can be easily generalized with the lemmas we developed in Section B.3.

Let us denote

$$\Psi(w_{\min}^\star, N) := (2 - 2^{\frac{N-2}{N}})^{\frac{1}{N-2}}(w_{\min}^\star)^{\frac{1}{N}} \wedge 2^{\frac{3}{N}}(2^{\frac{1}{N}}-1)^{\frac{1}{N-2}}(w_{\min}^\star)^{\frac{1}{N}}.$$

We set

$$\alpha \leq \left(\frac{\epsilon}{p+1}\right)^{4/N} \wedge \Phi(w_{\max}^\star, w_{\min}^\star, \epsilon, N) \wedge \Psi(w_{\min}^\star, N).$$

Under the same requirements on other parameters with Theorem 3, we satisfy the conditions of Lemma 4, Lemma 5 and Lemma 16. From these lemmas, we could maintain that

$$w_j^\star > 0 \implies 0 \leq w_t^- \leq \alpha^{N/4},$$
$$w_j^\star < 0 \implies 0 \leq w_t^+ \leq \alpha^{N/4},$$

up to $T_2 + T_4$ as defined in Proposition 1 and 2.

Consequently, for $w_j^\star > 0$ we can ignore $(w_{j,t}^-)_{t\geq 0}$ by treating as a part of bounded error $b_t$. The same holds for sequence $(w_{j,t}^+)_{t\geq 0}$ when $w_j^\star < 0$. Then, for $w_j^\star > 0$ the sequence $(w_{j,t}^+)$ evolves as follows

$$w_{j,t+1}^+ = w_{j,t}^+(1 - 2N\eta(w_{j,t}^+ - w_j^\star + (b_{j,t} - w_{j,t}^-) + p_{j,t})(w_{j,t}^+)^{(N-2)/2})^N.$$

The $b_{j,t} - w_{j,t}^-$ explains why we need $\|\mathbf{b}_t\|_\infty + \alpha^{N/4} \leq C_b\zeta$ in Proposition 1. For $w_j^\star > 0$, we follow the exact proof structure with Theorem 3 with treating $(w_{j,t}^-)_{t\geq 0}$ as a part of bounded error. For $w_j^\star < 0$ it follows the same argument by switching $w_t^+$ and $w_t^-$.

Therefore, we could closely follow the proof of Theorem 3 to generalize the result from non-negative signals to general signals. The result remains unchanged as well as the number of iterations requirement in equation (17) and (18). With the choice of $C_b = \frac{1}{100}$ in the proof of Theorem 3, recall that

$$\zeta = \frac{1}{5}w_{\min}^\star \vee 200\left\|\frac{1}{n}\mathbf{X}^\mathsf{T}\boldsymbol{\xi}\right\|_\infty \vee 200\epsilon,$$

and define the indicator function with $A$ as the event $\{\frac{1}{5}w^\star_{\min} > 200\left\|\frac{1}{n}\mathbf{X}^\mathsf{T}\boldsymbol{\xi}\right\|_\infty \vee 200\epsilon\}$,

$$\mathbb{1}(A) = \begin{cases} 1, & \text{when} \quad \frac{1}{5}w^\star_{\min} > 200\left\|\frac{1}{n}\mathbf{X}^\mathsf{T}\boldsymbol{\xi}\right\|_\infty \vee 200\epsilon, \\ 0, & \text{when} \quad \frac{1}{5}w^\star_{\min} \leq 200\left\|\frac{1}{n}\mathbf{X}^\mathsf{T}\boldsymbol{\xi}\right\|_\infty \vee 200\epsilon. \end{cases}$$

We now define that

$$
\begin{aligned}
T_l(\mathbf{w}^\star,\alpha,N,\eta,\zeta,\epsilon) &:= \frac{75}{16\eta N^2\zeta^{(2N-2)/N}}\log\frac{|w^\star_{\max}-\alpha^N|}{\epsilon} + \frac{15}{8N(N-2)\eta\zeta\alpha^{(N-2)}} \\
&\quad + \frac{6}{\eta N^2(w^\star_{\min})^{(2N-2)/N}}\log\frac{w^\star_{\min}}{\epsilon}\mathbb{1}(A), \\
T_u(\mathbf{w}^\star,\alpha,N,\eta,\zeta,\epsilon) &:= \frac{5}{N(N-1)\eta\zeta}\left(\frac{1}{\alpha^{(N-2)}} - \frac{1}{\alpha^{(N-2)/2}}\right) \\
&\quad + \frac{25}{N(N-1)\eta w^\star_{\min}}\left(\frac{1}{\alpha^{(N-2)/2}} - \frac{1}{\alpha^{(N-2)/4}}\right)\mathbb{1}(A).
\end{aligned}
$$
(25)

The error bound (16) holds for any $t$ such that

$$T_l(\mathbf{w}^\star,\alpha,N,\eta,\zeta,\epsilon) \leq t \leq T_u(\mathbf{w}^\star,\alpha,N,\eta,\zeta,\epsilon).$$

The equation (23) and (24) ensure that it is not a null set.

Thus, we finish generalizing Theorem 3 to general signals with an extra requirement $\Psi(w^\star_{\min},N)$ on the initialization $\alpha$.

For the case $k = 0$, i.e., $\mathbf{w}^\star = \mathbf{0}$, we set $w^\star_{\min} = 0$ and

$$\alpha \leq \left(\frac{\epsilon}{p+1}\right)^{4/N}.$$

Conditioning on $\|\mathbf{e}_t\|_\infty \leq \alpha^{N/4}$, we still have that

$$\|\mathbf{b}_t\|_\infty + \alpha^{N/4} \leq p\alpha^{N/4} + \left\|\frac{1}{n}\mathbf{X}^\mathsf{T}\boldsymbol{\xi}\right\|_\infty + \alpha^{N/4} \leq 2\left(\left\|\frac{1}{n}\mathbf{X}^\mathsf{T}\boldsymbol{\xi}\right\|_\infty \vee \epsilon\right) \leq C_b\zeta \leq \frac{1}{40}\zeta.$$

Therefore, by Lemma 5, for $\eta \leq \frac{1}{N(N-1)\zeta\alpha^{(N-2)/2}}$, we ensure $\|\mathbf{e}_t\|_\infty \leq \alpha^{N/4}$ up to $\frac{5}{N(N-1)\eta\zeta}\left(\frac{1}{\alpha^{N-2}} - \frac{1}{\alpha^{(N-2)/2}}\right)$, which agrees to the definition of $T_u(\mathbf{w}^\star,\alpha,N,\eta,\zeta,\epsilon)$ in this case. $\qquad\square$

## D.2  Proof of Corollary 1

Since $\boldsymbol{\xi}$ is made of independent $\sigma^2$-sub-Gaussian entries, by Lemma 3 with probability $1 - 1/(8p^3)$ we have

$$\left\|\frac{1}{n}\mathbf{X}^\mathsf{T}\boldsymbol{\xi}\right\|_\infty \leq 2\sqrt{\frac{2\sigma^2\log(2p)}{n}}.$$

Hence, letting $\epsilon = 2\sqrt{\frac{2\sigma^2\log(2p)}{n}}$, we obtain that

$$\|\mathbf{w}_t - \mathbf{w}^\star\|_2^2 \lesssim \sum_{i\in S}\epsilon^2 + \sum_{i\notin S}\alpha^{N/2} \leq k\epsilon^2 + (p-k)\frac{\epsilon^2}{(p+1)^2} \lesssim \frac{k\sigma^2\log p}{n}.$$

$\qquad\square$

## D.3  Proof of Theorem 2

We now state Theorem 2 formally as below.

**Theorem 4.** *Let $T_1$, $T_2$, $T_3$ and $T_4$ be the number of iterations defined in Proposition 1 and Proposition 2. Suppose $\zeta \geq 1$, $w^\star_{\max} \geq 1$ and the initialization $\alpha \leq \exp(-5/3)$, fixing $\alpha$ and $\eta$ for all $N$, both $T_2 - T_1$ and $T_4 - T_3$ have a tight lower bound that is increasing as $N$ increases $(N > 2)$.*

*Proof.* We observe first that under the assumption $\zeta \geq 1$ and $w^\star_{\max} \geq 1$, $\frac{75}{16\eta N^2\zeta^{(2N-2)/N}}\log\frac{|w^\star_{\max}-w_0|}{\epsilon}$ and $T_3 = \frac{6}{\eta N^2(w^\star_{\max})^{(2N-2)/N}}\log\frac{w^\star_{\min}}{\epsilon}$ are decreasing as $N$ increases.

For the rest part of $T_2 - T_1$, we will be showing that a lower bound of that is increasing as $N$ increases. As $T_2 - T_1$ is by design a lower bound of the "true" early stopping window, the lower bound we get here is tight for $T_2 - T_1$ and is treated as equivalent to $T_2 - T_1$ to indicate the monotonicity of the "true" early stopping window.

$$\frac{5}{N(N-1)\eta\zeta}\left(\frac{1}{\alpha^{(N-2)}} - \frac{1}{\alpha^{(N-2)/2}}\right) - \frac{15}{8N(N-2)\eta\zeta\alpha^{(N-2)}}$$
$$\geq \frac{5}{4N(N-1)\eta\zeta}\left(\frac{1}{\alpha^{(N-2)}} - \frac{4}{\alpha^{(N-2)/2}}\right)$$

Denote
$$f(N) = \frac{1}{N(N-1)}\left(\frac{1}{\alpha^{(N-2)}} - \frac{4}{\alpha^{(N-2)/2}}\right).$$

Therefore,
$$f'(N) = \frac{-(2N-1)}{N^2(N-1)^2}\left(\frac{1}{\alpha^{(N-2)}} - \frac{4}{\alpha^{(N-2)/2}}\right)$$
$$+ \frac{1}{N(N-1)}(-\log\alpha)\left(\frac{1}{\alpha^{(N-2)}} - \frac{4}{2\alpha^{(N-2)/2}}\right)$$
$$= \frac{-(2N-1) - (N-1)N\log\alpha}{2N^2(N-1)^2}\left(\frac{1}{\alpha^{(N-2)}} - \frac{2}{\alpha^{(N-2)/2}}\right)$$
$$+ \frac{2N-1}{N^2(N-1)^2}\frac{2}{\alpha^{(N-2)/4}}$$

Note that the second term is always positive, we just need to show the first term is positive.
$$-(2N-1) - (N-1)N\log\alpha \geq 0,$$
$$\frac{1}{\alpha^{(N-2)}} - \frac{2}{\alpha^{(N-2)/2}} \geq 0,$$

which is satisfied when
$$\log\alpha \leq \min_{N\geq 3}\frac{(1-2N)}{N(N-1)} = \min_{N\geq 3}\left(\frac{1}{1-N} - \frac{1}{N}\right) = -\frac{5}{6}$$
$$\alpha^{(N-2)/2} \leq 1/2.$$

We can further derive that when $\alpha \leq \exp(-5/6)\wedge 1/4$, we have a lower bound of $T_2 - T_1$ is increasing as $N$ increases.

To show $T_4 - T_3$ is increasing as $N$ increases, we just need to show $T_4$ is increasing. It follows a similar proof.

We can further derive that when $\alpha \leq \exp(-5/3)\wedge 2$, we have $T_4 - T_3$ is increasing as $N$ increases.

$\square$

### D.4 Proof of Remark 2

The proof is indeed similar to that of Theorem 4. Fixing any $N > 2$ and step size $\eta$, we look at $T_2 - T_1$ and $T4 - T3$ and show that a tight lower bound of that is increasing as $\alpha$ decreases. We start with $T_2 - T_1$.

Recall that

$$T_2 - T_1 = \frac{5}{N(N-1)\eta\zeta} \left( \frac{1}{\alpha^{(N-2)}} - \frac{1}{\alpha^{(N-2)/2}} \right) - \frac{15}{8N(N-2)\eta\zeta\alpha^{(N-2)}} )$$

$$- \frac{75}{16\eta N^2\zeta^{(2N-2)/N}} \log \frac{|w^\star_{\max} - \alpha^N|}{\epsilon}$$

$$\geq \frac{5}{N(N-1)\eta\zeta} \left( \frac{1}{\alpha^{(N-2)}} - \frac{1}{\alpha^{(N-2)/2}} \right) - \frac{75}{16\eta N^2\zeta^{(2N-2)/N}} \log \frac{|w^\star_{\max}|}{\epsilon}$$

Notice that the second term is not about $\alpha$. We just need to show that $f(\alpha) = \frac{1}{\alpha^{(N-2)}} - \frac{1}{\alpha^{(N-2)/2}}$ is increasing as $\alpha$ decreases. With the general requirement of $\alpha < 1$, we have that

$$f'(\alpha) = -\frac{(N-2)}{\alpha^{(N-1)}} + \frac{(N-2)/2}{\alpha^{N/2}}$$

$$= (N-2) \left( \frac{1}{2\alpha^{N/2}} - \frac{1}{\alpha^{(N-1)}} \right)$$

$$= (N-2)\frac{\alpha^{(N-2)/2} - 2}{2\alpha^{(N-1)}} < 0.$$

For $T_4 - T_3$, it follows a similar proof.

# E    Experiments on MNIST

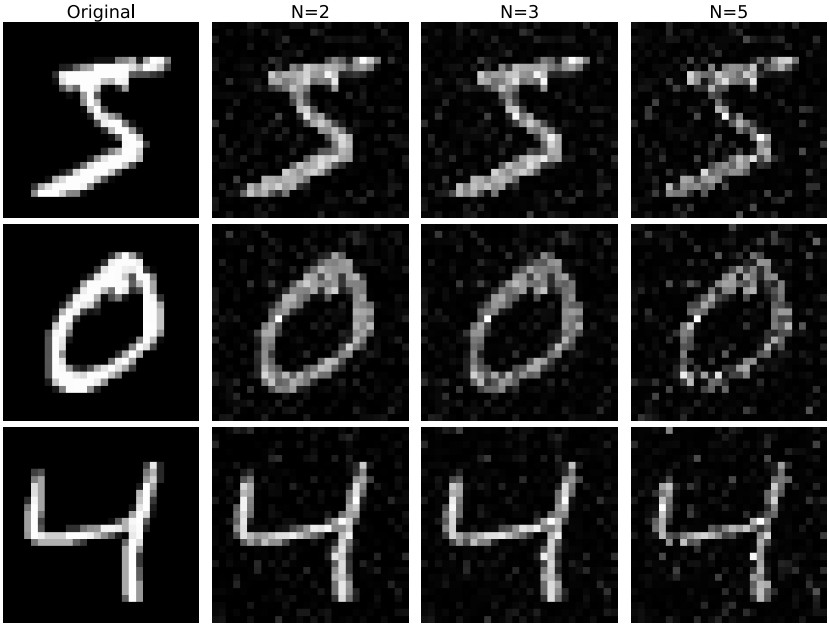

Figure 7: Experiments with different choice depth parameter $N$. The number of measurements is set as $n = 392$, where the dimension of the original image is $p = 784$. We use Rademacher sensing matrix.

The efficacy of different depth parameter $N$ is shown in Figure 2 and Figure 7 on both simulated data and real world datasets. The MNIST examples are successfully recovered from Rademacher linear measurements using different deep parametrizations.