# OpenReview forum: "Implicit Sparse Regularization: The Impact of Depth and Early Stopping"
_NeurIPS.cc/2021/Conference — NeurIPS 2021 Poster_

### Official Review · Reviewer_JHGS · 2021-07-04

**Rating:** 6
**Confidence:** 4

**Summary:**

In this paper the authors study implicit sparse regularization for diagonal linear networks. Specifically, the authors extend the previous result of Vaskevicius et al. [9] to the case of depth>2, showing that for noisy regression under a relaxed assumption on the inputs, increasing depth enlarges the scale of working initialization and the early-stopping window.

**Main Review:**

The paper is interesting and well written. However, my main concern is that this paper is quite similar to Vaskevicius et al. [9]. So, it would be good to add a detailed comparison to [9], e.g. state their bound and compare to the bound in this paper.

In addition, the empirical part is rather weak. I think it would be good to show empirically that the conclusions in this paper for the simple model hold also in more realistic deep networks trained on real data.

Comments:
- It would be good to define “minimax optimal recovery” and explicitly show that the bound is indeed minimax optimal.
- lines 65-66: “Woodworth et al. [16] study the gradient flow of the squared-error loss function in the noiseless case” – I think their result is rather general (noiseless or noisy). For any dataset (X,Y), taking $\alpha\rightarrow 0$ results in a minimum L1-norm solution.
 Also in line 184 “but did not realize the need of early stopping due to an oversimplification in the model”. Again, [16] discuss the implicit bias at convergence to global minima, so early stopping is irrelevant there.
- It would be good to discuss special cases in Theorem 1 – the noiseless case and the case $k=p$. Specifically, it seems that in the noiseless case $T_l$ and $T_u$ are undefined.
- I think the following two papers are relevant for discussion:

https://arxiv.org/abs/1903.11680

https://arxiv.org/abs/1903.09139
- When comparing the window size for different $\alpha$ (theorem 2 and Figure 1), I think that need to fix $\alpha^N$ and not $\alpha$, since $\alpha^N$ is the initialization scale of the model for any $N$.
- Can you explain Eq. (7) ? Why $\eta$ appears there, and did you use expectation of the loss over $X$, like in [11] ?
- In Figures 1 and 4, please explain how did you determine the location of the dashed black lines of the window.
- There is an unclear discrepancy between Figure 1 and Figure 4, since Figure 1 uses $\alpha$ but Figure 4 uses $\alpha^N$. I believe both figures should use $\alpha^N$ and show similar window size.
- Please explain why in Figures 1,4,5 the step-size is fixed for all $N$ but in  Figures 2,3 it is ~$\frac{1}{N^2}$.
- Would be good to add a discussion of $\mu$-coherence in simulations and the effect of the value of $\mu$. Specifically, does $\mu$-coherence hold for the $X$ used in simulations ?
- It is not clear what is the point of Figure 5 ? Incremental learning is not discussed/proved in this paper, and Figure 5 is just the same as the figures in [11].
- The discussion about kernel regime (Figure 6) seems detached from the paper. The relation between early stopping and ridge regression is well established in prior work, e.g.:

https://arxiv.org/abs/1903.11680

Typos:
- line 10: "an" in not needed
- line 179: third term -> second term
- line 240: converges -> converge
- line 270: Figure 4 -> Figure 1
- line 274: We -> we
- In Eq. (7): dL/dw_i should be dL/du_i.


**Time Spent Reviewing:**

15

---

> ### Author Response · Authors · 2021-08-11
> **Response to Reviewer JHGS**
>
> We appreciate your thorough review of our paper. We will try to address your main concerns and questions here.
>
> **Comparison to Vaskevicius et al.**
>
> Our work is indeed an extension of Vaskevicius et al. However, there are key differences. Let us highlight them.
>
> First, we note that our error bound for $N>2$ is the same as that in Vaskevicius et al. ($N=2$). With the sub-Gaussian noise, both works achieve a $\sqrt{\sigma^2 \log p /n}$ error rate, which is minimax-optimal. As for the requirement on hyper-parameters, both choices of initialization $\alpha^N$ ($\alpha^2$ in their case) scales with $\frac{1}{p^4} \wedge w_{max}^2 \wedge w_{min}$. The difference would be on the step size: in our case, the step size scales with $\eta \leq \frac{\alpha^N}{N^2}$ which depends on the initialization, where the choice in [9] doesn’t. This is because the gradient dynamics for general depth-$N$ settings is considerably harder to analyze (the corresponding recurrence relation is not solvable as pointed out by [11]; see page 2 of our paper for an explanation).  We propose a continuous approximation to handle this, and the extra requirement on the step size is needed to control the approximation error.
>
> Next, since Vaskevicius et al. only focus on a single $N$ (i.e., $N=2$), their result cannot be used to understand how the gradient dynamics changes with $N$. For example, our results show that the step size should scale with $1/N^2$. Larger depth-$N$ allows for larger initialization, and leads to larger early stopping window size. Those insights are not within the scope of [9].
>
> **Real world experiments.**
>
> As per your suggestion, we have performed sparse recovery experiments on a real-world dataset (MNIST). demonstrate the effectiveness of general depth-$N$; we are able to successfully recover various MNIST image instances (with p = 784) from Gaussian linear measurements.
>
>
> **Define “minimax optimal recovery” and explicitly show that the bound is indeed minimax optimal.**
>
> Minimax optimal recovery refers to the condition that the convergence rate of the error bound of $w$ achieves minimax optimal order. The minimax rate of linear models with sub-Gaussian noise can be found in various papers [e.g., BRT, SV]. Our result is minimax-optimal as indicated in Corollary 1.
>
> [BRT] P. Bickel, Y. Ritov, and A. Tsybakov. Simultaneous analysis of Lasso and Dantzig selector. Annals of Statistics, 2009.
>
> [SV] S. A. Van De Geer. High-dimensional generalized linear models and the Lasso. Annals of Statistics, 2008.
>
> **Discussion about Woodworth et al.**
>
> We agree that Woodworth doesn’t require noiselessness. We will modify our description related to Woodworth et al. in the revision.
>
> **Discussion about the noiseless case and the special case of $k=p$.**
>
> In the noiseless case with uncorrelated designs, the problem degenerates to the toy example in [11], where we admit $T_l$ and $T_u$ is undefined as early stopping is not needed. In either case (noisy or correlated designs), $T_l$ and $T_u$ is well defined.
> When $k=p$, we can still derive a lower bound of iteration number to ensure the convergence. In this work, we are more interested in the sparse setting where we assume $k\ll p$.
>
> **Two additional references.**
>
> Thank you for pointing these relevant references. We will provide additional discussion in the revision.
>
> **Fix $\alpha^N$ instead of $\alpha$ for theorem 2 and figure 1.**
>
> Our theory is built around fixing $\alpha$, which is a natural choice of the initialization to a diagonal neural network. The experiment result is shown in Figure 1. We agree that one can also vary $\alpha$ and $\eta$ with $N$ to conduct the analysis, depending on which specific implementations of gradient descent that one is interested in. We also showed that fixing $\alpha^N$ works experimentally in Figure 4.
>
> **Explanation of Eq (7).**
>
> First, the appearance of $\eta$ here is a typo; sorry for that. We just wanted to demonstrate different gradient dynamics compared with $N=2$. We do use the expectation of the loss over $X$ like [11]. As we pointed out, this equation just provides a simplified analysis and points out the technical difficulty even with this simpler form.
>
> **Dashed black lines Figures 1 and 4.**
>
> We know the required upper bound is $\alpha^{N/4}$ from Theorem 1 and also the true signal strength is 1 in our experimental setting. Therefore, the dashed black line is determined by when the maximum of entries outside the support crosses the upper bound and the smallest signal approaches 1 (with a tolerance parameter 0.1).
>
> **Discrepancy between Figure 1 and 4.**
>
> Our theoretical result is for fixed $\alpha$, which corresponds to Figure 1. Numerically, as $N$ increases, $\alpha^N$ would decrease dramatically and it would take a very large number of epochs to train (millions in Figure 1), which causes some difficulty to compute the “error-bar” in Figure 4. Therefore we provided an additional set of experiments with fixing $\alpha^N$ in Figure 4. Interestingly, the conclusions are similar to Figure 1. We apologize for the confusion, and will specify clearly that Figure 1 corresponds to our theoretical results and describe Figure 4 as a separate result.
>
> **Please explain why in Figures 1,4,5 the step-size is fixed for all $N$ but in Figures 2,3 it is $\frac{1}{N^2}$.**
>
> As the theory states, the step size should be with the scale of $\frac{1}{N^2}$. The reason that we choose constant step size for Figure 1,4,5 is that we want to illustrate the window increases with $N$. Choosing constant step size actually leads to a non-favorable scenario for fair comparison (to make sure that larger window size is not caused by smaller step size).
>
> **Add a discussion of $\mu$-coherence in simulations and the effect of the value of $\mu$. Specifically, does $\mu$-coherence hold for the $\mu$ used in simulations.**
>
> Sub-Gaussian random matrices satisfy low coherence with high probability [V]. We use the Radamacher random matrix in our experiments, and one can check that it indeed satisfies low coherence. The key reason we have use incoherence in our results instead of RIP is that it can be numerically matched to what our analysis needs. For example, we have verified that the incoherence of one instance in Figure 1 is 0.2480, which is smaller than our assumption of $1/\sqrt{5}$.
>
> [V] R. Vershynin, High-dimensional probability: An introduction with applications in data science, Cambridge University Press, 2018.
>
> **The point of Figure 5 and it is the same as the figures in [11].**
>
> Figure 5 is done in a general sparse recovery setting with correlated designs and the presence of noise, where the figure in [11] is based on a toy example. We both studied the gradient dynamics of depth-$N$ diagonal neural networks. Since [11] work on a much simplified setting, they obtain a stronger guarantee about the distinct phases of learning (incremental learning). Our result about early stopping windows is in fact a consequence of incremental learning, which seems to be an important phenomenon regarding depth-$N$ networks. We can experimentally show the incremental learning dynamics in the sparse recovery setting and will include further discussion on this point.
>
>  **Kernel Regime (Figure 6) seems detached from the paper.**
>
> Our high level motivation is to understand the gradient dynamics of (diagonal) depth-$N$ neural networks. Woodworth et al point out there are two regimes (rich v.s. kernel) with different scales of initialization. We theoretically showed and experimentally verified the rich regime of general depth-$N$ with small initialization. However, the precise gradient dynamics of depth-$N$ with larger initialization is still unclear. We provide Figure 6 to show that the kernel regime still remains with general depth-$N$ experimentally and early stopping plays a crucial role there; backing this up with theoretical guarantees is an interesting direction.
>
> We will include a discussion to state some phenomena experimentally shown and not theoretically proved (Figure 5 and 6).
>
> We have fixed all the typos pointed out by the reviewer. We hope this response would help to answer your concerns.

---

> > ### Comment · Reviewer_JHGS · 2021-08-21
> > **alpha vs alpha^N**
> >
> > Thank you for the feedback, I would like to increase my score by one point.
> >
> > I still think that it is more natural to fix $\alpha^N$ and not $\alpha$ when comparing different values of depth $N$, because $\alpha^N$ is the "equivalent" initialization scale of the output for any $N$. Note that $\alpha^N$ also appears in Theorem 1, e.g. in lines 144, 146, and the condition $\alpha\leq (..)^{4/N}$ can be written as $\alpha^N\leq (..)^4$.

---

### Official Review · Reviewer_LYJJ · 2021-07-13

**Rating:** 7
**Confidence:** 4

**Summary:**

This paper studies the implicit bias of gradient descent on the depth-N diagonal linear network for the problem of sparse recovery. It shows the convergence results for gradient descent and provides error bounds to the ground truth sparse signal. The result highlights the importance of early stopping in preventing overfitting and shows the role of network depth in extending the stopping window.

**Limitations And Societal Impact:**

Please refer to the comments.

**Main Review:**

This paper is well-written. I did not go through all the proofs but the proof ideas are clear to me. This work makes contributions in studying the deep network model (N>2) with more realistic settings: gradient descent, the presence of noise. Therefore I recommend “accept”.

Comments:

1. It seems to me that Theorem 1 does not apply to the case of N=2, since $\Phi$ and $\Psi$ are undefined. It would be great if the author can improve the theorem by including the case of N=2. Also, in line 149, it states that “the overall bound generalizes that of [9] for N=2”, I don’t see this as a direct generalization because [9] assumes RIP and this paper assumes $ \mu$-coherence.


2. For the initialization, u_0 and v_0 are initialized to be all one vectors scaled by small $\alpha$. Can we have similar implicit bias if one randomly initializes the entries of u_0 and v_0 with a small variance?


3. In other models where implicit bias arises under small initialization (Gidel’19), the initialization scale affects the time windows for early stopping. The author may want to discuss the effect of $\alpha$ on the time window.


Minor comment: line 140, the upper bound for $\mu$ is 1/kr, what is r?

Reference:

Gauthier Gidel, Francis Bach, and Simon Lacoste-Julien. Implicit regularization of discrete gradient
dynamics in linear neural networks. arXiv preprint arXiv:1904.13262, 2019.

**Time Spent Reviewing:**

10

---

> ### Author Response · Authors · 2021-08-10
> **Response to Reviewer LYJJ**
>
> We greatly appreciate the effort you have spent on reviewing our paper.
>
> **The result doesn’t apply to $N=2$.**
>
> You are correct; instead of “generalization”, our work can be viewed more as an extension of [9], and complements the results in [9] for depth-2 networks to the case of $(N>2)$. The use of RIP or incoherence actually does not lead to much difference in our analysis since [9] already implicitly uses incoherence in $\ell_\infty$-norm by applying Lemma A.3 in [9]. There is no difficulty in re-deriving our results using the RIP assumption. We use the incoherence condition because it is a computationally tractable measure, whereas verifying RIP for deterministically constructed design matrices is NP-hard [BDMS13].
>
> The major reason that we do not include the setting of $N=2$ is because the gradient dynamics at $N=2$ is degenerate, as seen in Equation (7). The presence of $w_i^{2-2/N}$ when $N>2$ in fact leads to major technical challenges, and we propose a continuous/first-order approximation approach to handle them. When $N=2$, this term disappears and no approximation is needed. It is therefore more suitable to directly quote the result for $N=2$ directly from [9].
>
>
> **Random initialization instead.**
>
> Yes, you are correct! Our results carry over if the entries of $u_0,v_0$ are randomly initialized, and the random variables concentrate in a small positive region (bounded away from 0), likely with an additional high probability assumption. We will include this observation in our revision.
>
>
>
> **The effect of $\alpha$ on the time window.**
>
> Based on our characterization of the time window, if we fix $N$ and $\eta$ to study the role of $\alpha$, we obtain a similar result (larger window size) as Gidel et al. Our result holds for general $N>2$, which can be viewed as an extension of their work to deeper linear neural networks. Thank you for bringing up this particularly interesting point. We will include more discussion on that in our revision.
>
> **Definition of $r$.**
>
> $r$ is defined in line 136, denoted as the condition number.
>
> Thank you again for your review, and we hope our response answers your questions.
>
> [BDMS13] Bandeira, Dobriban, Mixon, Sawin, Certifying the RIP is hard, IEEE Transactions on Information Theory, 2013.

---

### Official Review · Reviewer_P9DX · 2021-07-15

**Rating:** 6
**Confidence:** 4

**Summary:**

The paper under review studies the implicit sparse regularization induced by gradient descent on a diagonal linear network-like of arbitrary depth. Doing so, they extend a past work on depth 2 showing the possible benefit of larger depth. More precisely the authors show the role of early stopping to recover a sparse signal.

**Limitations And Societal Impact:**

Yes

**Main Review:**

**Overall comment and impression**

First, I would like to say that, despite its apparent technicality, the paper is clearly written and properly referenced. The reading is pleasant and understanding the results is almost effortless.

Second, the authors take the time to clarify the main ideas of the proofs and the overall dynamics taken by the gradient descent. This is, of course, highly appreciated.

Third, the main result is clear and answers (at least partially) to the non-trivial question of signal recovery in this depth N non-convex setting.

However, I have read in the past the article on depth 2 (referred as [9] in the paper) and was impressed by this result: this paper was one of the first on the subject and it compared well to other technique for sparse recovery. Even, if this paper extends its result to the N-depth case, I feel that the results are naturally *weaker* than the aforementioned paper. I do not doubt of the technical difficulty of extending the result to the N-depth case, but, despite this potentially considerable achievement, I do not see the real implications of such a result. More precisely, here is a list of concerns regarding the comparison with [9]:

- *Assumption on the data*. The assumption on the coherence of the matrix $\textbf{X}$ seems quite restrictive compared to the RIP property that seems largely adopted for sparse problem as a necessary and sufficient condition for recovery. Is there a link between RIP and coherence ? Can the authors provide simple settings (e.g. Gaussian variable) for which they have a control on $\mu$ ? How the hypothesis $\mu < 1/kr$ can limit the applicability of the Theorem ? What about lower bounds with this assumption ? The reference [29] given to justify that the assumption is "now classical" dates back to 2005... In a word, I would expect more justifications.

- *A not-so-clear benefit from depth N*. For me the benefit of depth N diagonal networks over depth 2 case really need to be clarified. Indeed, with the current justification of the authors (mainly Remark 1 and Theorem 2), I have hard times to conclude. More precisely,
  - The *stability* perspective offered by Theorem 2 is not very clear. First, the authors only gave a lower bound on the time-window, and it is not really clear whether this estimate is tight (see argument on *Precise role of early stopping* below). Going further, they fixed $\alpha$ and $\eta$ to state Theorem 2 and, as they both depend on N, it is very not clear whether large N enlarges the time window. Finally, it is known (and shown in Theorem 1) that, concerning the step sizes, the larger the N, the less stable the gradient descent is! Hence, I am very confused with this statement.
  - When comparing the initialization scales of depth 2 and depth N, it must really be clear that the *Optimization speed* are going to be different! At fixed depth, the larger the initialization, the faster the convergence. Yet, the deeper the diagonal neural nets, the slower the escape from $0$. As clearly pictured in figure 2, depth 2 clearly wins the trade-off. This is one of the reason, it is difficult to conclude for me that depth is preferable for these systems (and I know that the authors do not conclude this crudely).

Beyond the comparison with [9], here is a list of questions (that can, I admit, be also applicable to the $N=2$ case):
- *Tightness of initialization and step sizes*. I really would like a discussion on the tightness with respect to the initialization and step size, that both go with the inverse of the dimension. Is this really needed ? Do the authors observe this in practice ? Because this can slow down dramatically the dynamics in high dimension.
- *Precise role of early stopping*. I am not clear why early stopping is necessary. When there is no noise, we know from papers the authors cited that it is not the case. But I really wonder if the noise really ruins the results (let us even assume that it is Gaussian). Somehow I feel like the noise can make the iterates fluctuate but may not totally blow the iterates away. Maybe a tighter control of this part of the decomposition could lead to a more in depth analysis of the dynamics.

**Minor comments**

- Line 86, I guess that n should be k.
- Line 105, can the authors develop a bit on the fact that the noisy setting should be more realistic and lead to more insights ? This is quite the opposite of what the current machine learning community states...
- Comment on the fact that $u_0 = v_0 = \alpha$, what about usual gaussian initialization used in practice?
- I would like scales of $T_u$ and $T_l$ appear in the main text as they are important quantities.

**Conclusion**

I would be happy to increase my score if the authors manage to convince me that the depth N model gives new insights on recovering a sparse signal or on the  training of neural networks. However, even if the technical achievements of the paper are undeniable, I would not accept the paper under the current form.

**Time Spent Reviewing:**

5

---

> ### Author Response · Authors · 2021-08-11
> **Response to Reviewer P9DX**
>
> Thank you for the detailed review of our paper.
>
> **Discussion about the data assumption (incoherence vs. RIP).**
>
> Both incoherence and RIP have been widely used for establishing various sparse recovery guarantees, and the connection between them has been well-studied [CENP, CR, DE]. Subgaussian random matrices are known to satisfy low-incoherence with high probability [V]. We use the incoherence condition because it is a computationally tractable property that can be verified in polynomial time; on the other hand, verifying the RIP for any given design matrix is NP-hard [BDMS]. We see no technical difficulty in re-deriving our results with the RIP assumption similarly to [9].
>
> **The stability perspective offered by Theorem 2 is not very clear.**
>
> We only mention the word “stable/stability” in the paper abstract; we admit in hindsight that it was not a good word choice and might have created confusion. With Theorem 2, we provide guarantees on the signal recovery within the early stopping window (rather than claiming any particular notion of algorithmic stability). As for the choice of $\alpha$ and $\eta$, their dependence on $N$ are merely in terms of the upper bounds in Theorem 1. For comparison between two different values of $N$, it is valid to fix $\alpha$ and $\eta$ small enough to fulfill the upper bounds required by both values of $N$. We agree that one can also vary $\alpha$ and $\eta$ with $N$ to conduct the analysis, depending on which specific implementation of gradient descent is used. Another possibility is to choose $\alpha$ and $\eta$ as the maximum values allowed in Theorem 1. However, we are not aware if the upper bounds (especially that of $\alpha$) are tight. Therefore, we choose one of the most straightforward settings -- fixing $\alpha$ and $\eta$ in Theorem 2.
>
> **Lower bound of the time window.**
>
> As we also commented in the paper, the estimate of the time window is a lower bound. Obtaining the precise bound is challenging as the recurrence relation associated with the gradient descent for general depth-$N$ is not solvable, which has also been previously pointed out by Gissin et al., [11]. That said, if we let the step size go to 0, the lower bound we developed actually converges to the true window size, as we use a first-order approximation idea to overcome the technical difficulties. Our experiments confirm that our characterization of the time window is quite accurate, see Figures 1 and 4.
>
> **Benefit from depth-$N$.**
>
> We would like to emphasize that we do not intend to claim a clear benefit of depth-$N$ versus depth-$2$. Rather, our goal is to understand the effect of the depth in terms of implicit regularization, and to provide insights into the training of neural nets with depth >2. There has been a growing interest in theoretical understanding of deep neural network training, and our work provides a thorough analysis for the special case of training diagonal linear networks. An important consequence of our analysis is that the early stopping window becomes wider as $N$ increases. Our theoretical result towards larger initializations and early stopping windows may help shed light on the question about why, despite being massively overparameterized,  deeper neural networks that perform well can be reliably trained.
>
> **Tightness of initialization and step sizes.**
>
> The dependency of step size on the dimensions comes from the initialization. As for the initialization, Vaskevicius et al [9] suffers from a similar problem. Since we both want to bound the accumulated error outside the support in $\ell_\infty$ from (see Lemma 1). For our experiments, the scale of initialization and step size ranges from $10^{-3}$ to $10^{-6}$, while $(\frac{1}{p})^4$ scales around $10^{-11}$. This might suggest those bounds are not tight enough, though smaller initialization and step size always leads to better results.
>
> **Precise role of early stopping.**
>
> We do not claim that the solution diverges, or that the iterates are blown away. Although we do not have a formal theoretical justification, our empirical experiments (see Figures 1 and 4) suggest that the quality of the solution deteriorates after the early stopping window, and at least remains worse even after a significant number of iterations. Even if it fluctuates back eventually, it seems to require a very, very large number of iterations (at least in the numerical examples we have tried so far), which would already make early stopping an attractive strategy. In other words, the empirical evidence suggests that the iterations do transition out of the desired window eventually (if we define “desired” as the case where all the nonzeros of the underlying signal are accurately recovered) and therefore early stopping is sufficient to avoid this event. Whether or not it is *necessary* is an interesting open question.
>
>
> **Minor comments.**
>
> Regarding the Gaussian initialization instead, if we think of the random initialization that concentrates on a small positive region (distinct from 0), our analysis still applies (with modification to a probabilistic statement).
>
> In the revision we will try to make $T_u$ and $T_l$ appear in the main context without adding unnecessarily sophisticated terms.
>
> We really appreciate your efforts in reviewing our work; it made us reflect upon several new directions. We hope this response helps to address your concerns.
>
> [CENP] Compressed sensing with coherent and redundant dictionaries, Applied and Computational Harmonic Analysis, 2011.
>
> [CR] E. Candes and J. Romberg, Sparsity and incoherence in compressive sampling, Inverse Problems, 2007.
>
> [DE] D. Donoho and M. Elad, Optimally sparse representation in general (nonorthogonal) dictionaries via $\ell_1$ minimization, PNAS, 2003.
>
> [V] R. Vershynin, High-dimensional probability: An introduction with applications in data science, Cambridge University Press, 2018.
>
> [BDMS] Bandeira, Dobriban, Mixon, Sawin, Certifying the RIP is hard, IEEE Transactions on Information Theory, 2013.

---

> > ### Comment · Reviewer_P9DX · 2021-08-26
> > **Final comment**
> >
> > I thank the authors for their commitment into the paper.
> >
> > I keep on thinking that this early-stopping question is crucial to understand qualitatively what happens with noise in this type of non-convex optimisation dynamics! However, the clarity of the rebuttal added to the fact that my concern on incoherence vs RIP has totally vanished make me upgrade my score to 6.

---

### Official Review · Reviewer_WsVt · 2021-07-16

**Rating:** 7
**Confidence:** 4

**Summary:**

In this work, the authors study the implicit bias of gradient descent for sparse linear regression when parameterizing the ground truth signal as a depth-$N$ diagonal linear network. Previous works have shown that when utilizing a Hadamard parameterization $w = u \odot u - v \odot v$, gradient descent exhibits an implicit bias towards sparse solutions, without any explicit regularization encouraging sparsity. Follow-up work has also considered the depth-$N$ case in simplified settings (e.g., without noise, positive signals, and idealized measurement setups). The authors extend this analysis to the general depth-$N$ case $w = u^N - v^N$ in the presence of noise. They prove that when the measurement matrix is $\mu$-incoherent with sufficiently small $\mu$ and noise is present, there is an early stopping window depending on the problem parameters where any iterate during this time is close in $\ell_{\infty}$ norm to the underlying ground truth sparse signal. These results hold for any depth $N \geqslant 2$ and showcase that depth allows for larger initialization sizes and that the early stopping window grows with deeper parameterizations. Experimental evidence on toy examples is also provided corroborating several elements of the proposed theory.

**Limitations And Societal Impact:**

This work does not have a clear societal impact. The authors have discussed some limitations of their work, such as the scaling on the coherence parameter $\mu$. Some example cases for certain design matrices and how many measurements are required to satisfy $\mu \leqslant 1/kr$ for various $r$ and distributions on $X$ would be good to see. In terms of other limitations, a short discussion or mention of what other phenomena concerning depth has not been theoretically shown, for example the incremental learning dynamics verified experimentally but not proven.

**Main Review:**

# Originality

The present work appears to be a non-trivial extension of the results in [9] to the depth-$N$ case. While some of the analysis appears similar, some of the techniques in the quadratic case do not immediately port over to the depth-$N$ case and this work offers insights into the benefits of depth. On this note, I believe another work along these lines that should be cited is Li et al (ICLR 2021), who also showed in depth-$N$ matrix factorization, a deeper parameterization allowed for larger initialization sizes.

- Zhiyuan Li, Yuping Luo, and Keifeng Lyu. Towards Resolving the Implicit Bias of Gradient Descent for Matrix Factorization: Greedy Low-Rank Learning. ICLR 2021

# Quality

- Strengths: - The analysis presented is for general depth $N \geqslant 2$ and no restrictions on the sign of the ground truth signals entries. - The results apply to gradient descent, as opposed to usual simpler analyses for gradient flow. - The impact of depth is discussed in multiple ways in the results. In particular, the results show that “deeper” diagonal networks allow for larger initialization sizes to still succeed and that the early stopping window becomes larger with larger values of $N$. - Various aspects of the theory are experimentally shown in toy examples.

- Weaknesses: - It seems that the condition number $r = w_{\max}^*/w_{\min}^*$ is integral to the analysis, but there is little discussion about how it influences the results. For example, given a signal with $w_{\max}^* = O(1)$ and $w_{\min}^* = O(1/k)$, how would the generalization error in Corollary 1 scale? This would require a very tight incoherence parameter $\mu \leqslant 1/k^2$, so I wonder if for certain design matrices $X$ (e.g., subGaussian), this could result in sub-optimal sample complexity scalings.

# Clarity

Overall, the paper is well-written and previous work seems to be properly placed in the context of the results. The proof sketches and discussions on the theorem’s assumptions and implications were quite helpful in parsing the result.
- Typos: pg 4 line 136 “property of design” -> “property of the design”

# Significance

While the implicit bias for the quadratic case is known and well-studied, the extension to the depth-$N$ case appears non-trivial. It is good to know that such a result does hold for the depth-$N$ case and the result adds to the literature on the various benefits of depth (albeit in a simplified setting).

**Time Spent Reviewing:**

4

---

> ### Author Response · Authors · 2021-08-10
> **Response to Reviewer WsVt**
>
> We would like to thank you for the constructive feedback and review.
>
> **Missing reference.**
>
> We will cite and discuss the related work of Li et al. (2021) in the revision. Thank you for pointing out this reference.
>
> **The condition number and its impact on the incoherence parameter.**
>
> The condition number appears in the incoherence condition for the design matrix. Our requirement of the incoherence parameter is $\mu\leq\frac{1}{kr}$, which is somewhat worse than [9], which requires the RIP constant $\delta \leq \frac{1}{\sqrt{k} \log r}$. However, the depth-2 setting in [9] leads to far simpler gradient dynamics, since the term $w_t^{(N-2)/N}$ in Equation (6) degenerates to 1. To overcome this barrier for $N > 2$, we propose a new first-order/continuous approximation. The technical challenge of properly handling the approximation error (specifically, Equation (21) of the supplement) results in a stronger requirement on the incoherence parameter. On the one hand, asymptotic rates of recovery of the true signal are not significantly affected; as shown in Corollary 1, the rate of convergence is minimax-optimal and the same as [9]. On the other hand, as you mentioned, it does lead to a worse sample complexity scaling. See the next paragraph.
>
> **Sub-optimal sample complexity.**
>
> Although we did not discuss the sample complexity in detail, with some simple calculations one can see that our sample complexity scales as the order $k^2 r$, and that in [9] is $k^2 \log^2 r$. Neither our work nor [9] achieves the optimal sample complexity $k\log p/k$. However, we stress that our goal is not to achieve an optimal approach for standard sparse recovery, but rather to formally understand how the depth parameter in certain types of over-parameterizations affects implicit (sparse) regularization. We will expand on this discussion in the revision.
>
> **Other phenomena not theoretically shown.**
>
> Thank you for the constructive suggestion. We will summarize other phenomena that we have verified experimentally (but not proven) such as incremental learning and kernel regime, in the revision.
>
> We will fix the typo that you pointed out. Thanks!

---

> > ### Comment · Reviewer_WsVt · 2021-08-26
> > **Post-rebuttal response**
> >
> > Dear authors,
> >
> > Thank you for your detailed responses. After reading the other reviews and responses, I would like to increase my score by a point. Even if the sample complexity scaling is sub-optimal, I think the work helps add to our understanding of the influence of depth in this diagonal linear model.

---

### Decision · Program_Chairs · 2021-09-27

**Decision:**

Accept (Poster)

**Comment:**

In this paper the authors study the implicit bias of gradient descent for sparse linear regression problem. They focus on a particular parameterization where the model is the hadamard product of two vectors and more generally deep diagonal linear network. Prior literature has shown the propensity or implicit bias of gradient descent towards sparse solutions in some settings including deep linear cases but with simplified assumptions and no noise. The authors' result however applies in the general deep diagonal linear case and also considers noise. The authors show that under proper incoherence assumptions there is an early stopping window in which the reconstruction error is sufficiently small. The authors also utilize their theory to provide a variety of insights. The reviewers overall liked the generality of assumptions compared to prior work and thought the writing was clear. The reviewers did raise some concerns about (1) related work, (2) originality, (3) condition number of signal, (4) too strong of an incoherence assumption, (5) and lack of special insight for deep networks. During the discussion period the authors addressed some of the questions raised which led to reviewers raising the scores. My own reading of the paper is similar. The paper has some nice results but with a few caveats which limits the utility. Therefore, I recommended acceptance as a poster. I strongly urge the authors to revise the paper per reviewer suggestions.